# Cold exposure promotes the progression of osteoarthritis through downregulating APOE in cartilage

Yueqi Zhang [1,12], Mei Fu[1,2,12], Chun Zhou[3,12], Xiao Wang[4], Zengxin Jiang[5], Chang Jiang [6,7], Shengyang Guo[8], Zhiying Pang[9], Chenzhong Wang[6], Tao Yu[10], Senbo An [11✉], Xiuhui Wang [8✉] & Zhe Wang [3✉]

## Abstract

**Osteoarthritis (OA) is a degenerative joint disease with limited effective therapies. Cold weathers have been shown to affect joint pain in OA patients. However, the impact of cold climate on OA progression is debated, with the underlying mechanisms not fully understood. This study aims to elucidate the role of Apolipoprotein E (*Apoe*) in chondrocytes in relation to OA progression under cold exposure. Both human chondrocytes RNA sequencing and DMM mice OA model revealed that lower temperatures significantly downregulated *Apoe* expression, correlating with OA exacerbation. Conditional knockout of *Apoe* in cartilage aggravated cartilage degeneration, leading to lipid accumulation, increased ROS production, mitochondrial dysfunction, and elevated chondrocyte apoptosis. Treatment with RGX-104, an LXRβ agonist, reversely restored APOE expression, mitigated aberrant lipid accumulation and countered the detrimental effects of cold exposure on OA progression. These results suggest that targeting lipid transfer and metabolism, especially through *Apoe* modulation, may offer therapeutic strategies for OA patients residing in colder climates, such as those at high altitudes and latitudes, and even winter season.**

**Keywords** Apolipoprotein E; Cold Exposure; Lipid Accumulation; Osteoarthritis; ROS
**Subject Categories** Evolution & Ecology; Immunology; Musculoskeletal System

## Introduction

Osteoarthritis (OA), one of the most prevalent degenerative joint diseases, mainly affecting the weight bearing joint like knee, is the leading cause of joint pain, swelling and stiffness, which impairs the motor ability and quality of life of patients (Glyn-Jones et al, 2015). Patients with OA often remark that the weather has an impact on their level of discomfort (Laborde et al, 1986). Timmermans et al found that approximately two-thirds of OA patients were weather-sensitive, most of whom reported cold as one of the main factors affecting their pain (Timmermans et al, 2014). In another European study, joint pain was found to be stronger in cold weather conditions (Timmermans et al, 2015). It is yet unknown, how cold exposure and osteoarthritis are related. In this research, we explored how cold exposure affects OA and the underlying mechanism.

In mammals, to maintain body temperature during cold exposure, lipid metabolism is altered to promote adaptive thermogenesis by processes such as browning of white adipose tissue (Seale et al, 2011). Numerous research studies have indicated that exposure to cold induces modifications in the genes and metabolic pathways linked to lipid metabolism. Xu et al, found that the lipid composition of iWAT changes at low temperatures, with fatty acids, diglycerides, and cardiolipin significantly increasing (Xu et al, 2019). Moreover, genes related to the elongation of fatty acids and the synthesis of different types of lipids, including triacylglycerols and sphingolipids, are expressed differently in response to cold exposure (Xu et al, 2019). Apart from adipose tissue, cold exposure also modifies the gene expression related to lipid metabolism in other cell types. Ruixia et al found that hypothermia and hypoxia downregulated fatty acid metabolism factors such as irisin and UCP-1 in skeletal muscles (Ruixia et al, 2021). Lipid peroxidation was found to be provoked in cold stress-induced cardiovascular dysfunction (Yin et al, 2020). Moreover, cold

[1]Department of Traumatic Surgery, Shanghai East Hospital, School of Medicine, Tongji University, 200092 Shanghai, China. [2]Department of Emergency Medicine, Renji Hospital, School of Medicine, Shanghai Jiao Tong University, 200127 Shanghai, China. [3]Orthopaedic Trauma, Department of Orthopedics, Renji Hospital, School of Medicine, Shanghai Jiao Tong University, 200127 Shanghai, China. [4]Department of Orthopedic Surgery, School of Medicine, Johns Hopkins University, Baltimore, MD 21205, USA. [5]Department of Orthopedics, Shanghai Sixth People's Hospital, 200233 Shanghai, China. [6]Department of Orthopedics, Zhongshan Hospital, Fudan University, 200032 Shanghai, China. [7]Shanghai Clinical Research and Trials Center, ShanghaiTech University, 201210 Shanghai, China. [8]Department of Orthopedics, Shanghai University of Medicine & Health Sciences Affiliated to Zhoupu Hospital, 201318 Shanghai, China. [9]Department of Joint Surgery, Shanghai East Hospital, School of Medicine, Tongji University, 200092 Shanghai, China. [10]Department of Spine Surgery, Guangdong Provincial People's Hospital (Guangdong Academy of Medical Sciences), Southern Medical University, 510080 Guangzhou, China. [11]Department of Joint Surgery, Shandong Provincial Hospital Affiliated to Shandong First Medical University, 250021 Jinan, Shandong, China. [12]These authors contributed equally: Yueqi Zhang, Mei Fu, Chun Zhou. ✉E-mail: ansenbo@sdfmu.edu.cn; zpyygkwxh@sina.cn; wangzhe@renji.com

temperature increased the intracellular neutral lipid droplet content in muscle cells to enhance oxygen diffusion through the muscle tissue (Sidell, 1998). However, whether cold temperature alters lipid metabolism in osteoarthritic chondrocytes is still unclear.

Apolipoprotein E (APOE), mainly produced by the liver and distributed widely in the human body, is a transporter molecule essential for the elimination of VLDL, LDL, and chylomicrons (Edwards et al, 2002). Since it has been established that *Apoe* is involved in maintaining cholesterol homeostasis, *Apoe* knockdown (KO) in mice has emerged as a potent technique for inducing atherosclerosis in cardiovascular disease (Piedrahita et al, 1992). Serving as a low-density lipoprotein (LDLR) ligand with high affinity, APOE facilitates the internalization of lipids into hepatic and extrahepatic cells (Edwards et al, 2002; Piedrahita et al, 1992). As an LXR target gene, *Apoe* participates in the reverse transport of cholesterol (Ouimet et al, 2019). Over-accumulation of the intermediate products of cholesterol, oxysterol and desmosterol, stimulates liver X receptors (LXRs), which triggers the activation of genes like *Abca1* and *Abcg1* involved in cholesterol efflux. Then, excess lipids are transferred to HDL particles or apolipoproteins, such as APOA1 and APOE, by ABCA1 and ABCG1, to complete the elimination of excessive cellular lipids (Favari et al, 2009; Sankaranarayanan et al, 2009; Vedhachalam et al, 2007). Studies in the past have demonstrated that *Apoe* deficiency influences lipid metabolism and is linked to progression of OA. Saba Farnaghi et al discovered that *Apoe* KO mice given a high cholesterol diet (HCD) spontaneously changed to exhibit OA-like changes, and that DMM mice given an HCD showed more severe OA manifestations in comparison to CD mice (Farnaghi et al, 2017). In addition, De Munter et al found that compared with WT OA mice, *Apoe* KO OA mice displayed increased ectopic bone formation, primarily at the medial collateral ligament (de Munter et al, 2016). Further, comparing *Apoe* KO mice to WT mice, sustained and enhanced semichronic inflammatory arthritis was observed (Archer et al, 2016). Last but not least, a global analysis of nuclear receptor expression revealed downregulation of LXRα and LXRβ as well as LXR target genes such as *Apoe*, *Apod*, and *Abcg1*, when OA cartilage was compared to normal cartilage (Collins et al, 1965). These investigations showed a correlation between the severity of OA and *Apoe* deficiency. However, whether the cold temperature could alter the level of ApoE in chondrocytes and cartilage remains unclear.

Our study revealed that cold exposure exacerbates osteoarthritis by promoting lipid accumulation in chondrocytes and cartilage tissue. Mechanistically, this effect occurs through downregulation of *Apoe*, a key regulator of lipid metabolism, which subsequently enhances intracellular lipid accumulation and induces chondrocyte apoptosis.

# Results

## Cold exposure aggravates cartilage degeneration in the DMM mouse model

To explore the relationship between mice knee joint temperature and ambient temperature, knee joint temperatures of mice were measured at gradually decreasing ambient temperatures (Fig. 1A; Appendix Fig. S1). At an ambient temperature of 25 °C, the knee joint cavity temperature of the mice was at around 35.5 °C. The knee joint cavity temperature gradually decreased with a decrease in ambient temperature. When the ambient temperature fell to

about 10 °C, the knee joint cavity temperature dropped and maintained ~33 °C. However, as the ambient temperature continued decreasing to 5 °C and even to 0 °C, the decline in knee joint cavity temperature began to stabilize around 33 °C, showing little further reduction (Fig. 1B). It was observed that mice succumbed when the ambient temperature dropped further below freezing point. Accordingly, 10 °C was selected as the experimental ambient temperature for in vivo studies and 33 °C for in vitro cell experiments to simulate cold exposure.

Wild-type C57BL/6 mice undergoing DMM surgery were housed at either 10 °C or 23 °C for 8 weeks (Fig. 1I). Open field test indicated reduced total distance traveled by mice at lower temperatures, but no significant differences in other parameters (Fig. 1C–H). Cartilage lesion evaluation revealed more severe damage at low temperatures, supported by the smaller safranin O-stained area in the DMM group at low temperatures (Fig. 1J), despite the total distance traveled by the mice being less in the low temperature group. OARSI scores were higher in the DMM group at low temperatures compared to room temperature, indicating increased cartilage damage severity (Fig. 1M). COLII expression was reduced (Fig. 1K,N), while MMP13 expression was elevated (Fig. 1L,O) in the low-temperature group. qPCR analysis confirmed lower both *Col2a1* and *Acan* mRNA levels, and higher *Mmp13* and *Adamts5* levels in the LTDMM groups as opposed to RTDMM groups (Fig. 1P–S). These findings collectively suggest that cold temperatures may exacerbate cartilage damage severity, hinder anabolism, and promote catabolism of the cartilage matrix in the DMM mouse model.

## *Apoe* downregulation in cartilage correlates with cold exposure in the progression of OA

To investigate the mechanisms by which low temperature exacerbate OA, RNA sequencing on human chondrocytes cultured at 33 °C and 37 °C was performed. The gene expression in non-OA region chondrocytes and OA region chondrocytes at 33 °C was compared to that at 37 °C. Principal Component Analysis (PCA) revealed that temperature (33 °C or 37 °C) was the primary factor contributing to variance. Additionally, a significant genotype effect was observed between OA and control groups (Fig. EV1A). At 33 °C, the OA group had 89 downregulated and 130 upregulated genes compared to 37 °C (Fig. 2A). Functional analysis through Gene Ontology (GO) and Kyoto Encyclopedia of Genes and Genomes (KEGG) enrichment showed involvement of genes in cell adhesion, extracellular matrix organization, and lipid metabolism pathways (Fig. EV1B). Particularly, lipid metabolism pathways such as fatty acid biosynthesis and cholesterol metabolism were significantly affected (Fig. 2D). Further analysis found 197 differentially expressed genes between 33 and 37 °C non-OA chondrocytes (Fig. EV1C). Overlap analysis identified 68 genes shared between OA chondrocytes and non-OA chondrocytes in response to temperature change (Fig. 2B). Based on our functional analysis, we hypothesized that the exacerbating effect of low temperature on osteoarthritis is linked to lipid metabolism. Among the 68 temperature-relevant genes, *Apoe* was significantly downregulated at 33 °C (Figs. 2C and EV1D), suggesting its specific role in lipid metabolism.

Western blotting confirmed lower APOE protein expression in IL-1β-treated murine OA chondrocytes at 33 °C compared to 37 °C

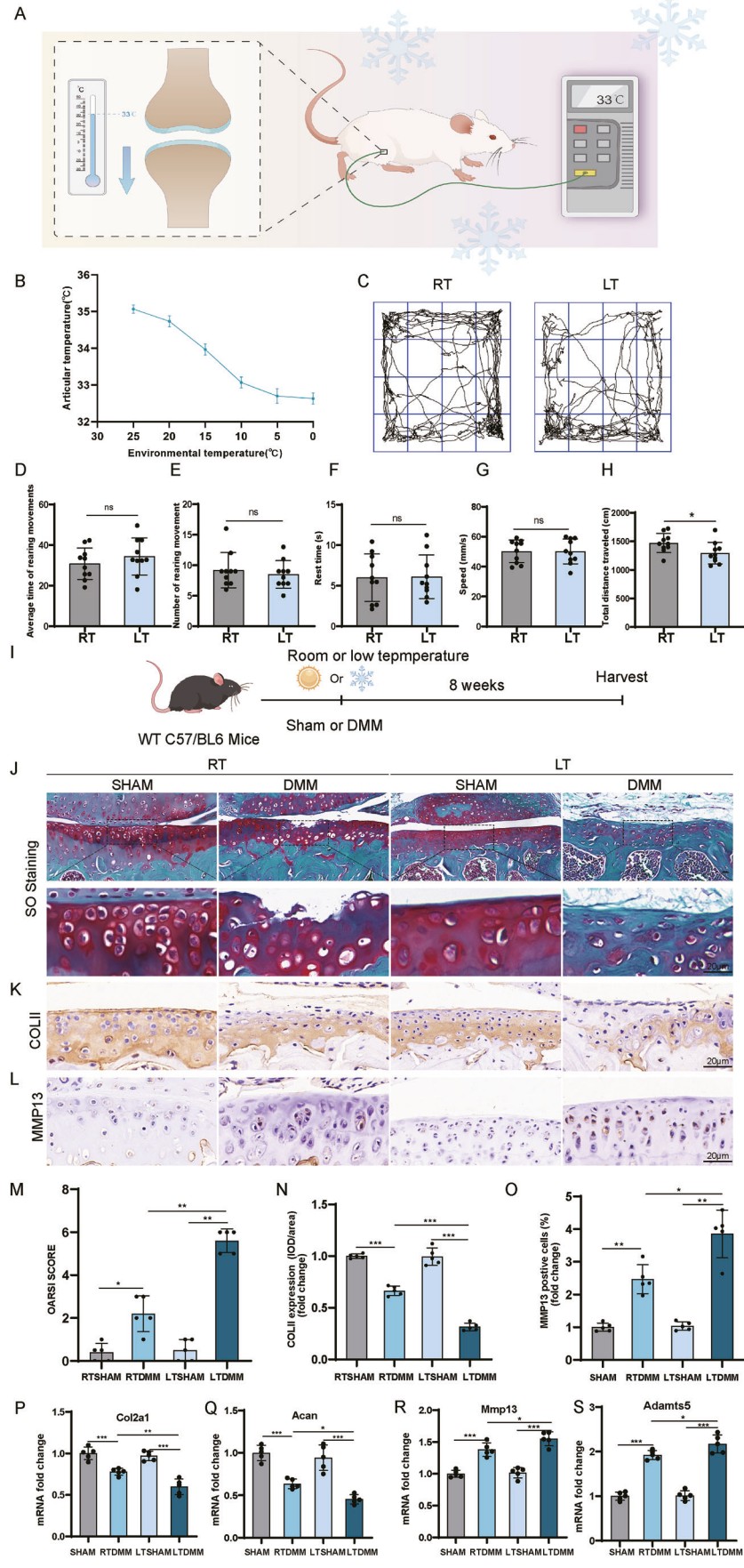

**Figure 1. Low temperature exacerbated the progression of OA in DMM mouse model.**

(A) Scheme of measurement of articular temperature of the mice. (B) Curve of articular temperature changing with ambient temperature ($n = 3$ for each ambient temperature). (C) Pathway image of the open test of mice at different temperatures. (D) Average time of rearing movements ($n = 10$). (E) Number of rearing movement ($n = 10$). (F) Rest time ($n = 10$). (G) Speed ($n = 10$). (H) Total distance traveled ($n = 10$). $P = 0.0401$. (I) Schematic illustration showing the design of DMM surgery of mice and exposure of room or low temperature. (J) Safranin O/Fast Green staining of the mice knee joints. (K) Immunohistochemistry staining of COLII in the mice knee joints. (L) Immunohistochemistry staining of MMP13 in the mice knee joints. (M) OARSI score of the mice knee joints ($n = 5$). $P = 0.0159$ (RTSHAM vs. RTDMM), $P = 0.0079$ (RTDMM vs. LTDMM), $P = 0.0079$ (LTSHAM vs. LTDMM). (N) Analysis of COLII level ($n = 5$). $P < 0.0001$ (RTSHAM vs. RTDMM), $P < 0.0001$ (RTDMM vs. LTDMM), $P < 0.0001$ (LTSHAM vs. LTDMM). (O) Analysis of MMP13 level ($n = 5$). $P = 0.0041$ (RTSHAM vs. RTDMM), $P = 0.0349$ (RTDMM vs. LTDMM), $P = 0.0028$ (LTSHAM vs. LTDMM). (P) mRNA fold change of Col2a1 ($n = 5$). $P = 0.0006$ (RTSHAM vs. RTDMM), $P = 0.0039$ (RTDMM vs. LTDMM), $P < 0.0001$ (LTSHAM vs. LTDMM). (Q) mRNA fold change of Acan ($n = 5$). $P < 0.0001$ (RTSHAM vs. RTDMM), $P = 0.0376$ (RTDMM vs. LTDMM), $P < 0.0001$ (LTSHAM vs. LTDMM). (R) mRNA fold change of Mmp13 ($n = 5$). $P < 0.0001$ (RTSHAM vs. RTDMM), $P = 0.0423$ (RTDMM vs. LTDMM), $P < 0.0001$ (LTSHAM vs. LTDMM). (S) mRNA fold change of Adamts5 ($n = 5$). $P < 0.0001$ (RTSHAM vs. RTDMM), $P = 0.0467$ (RTDMM vs. LTDMM), $P < 0.0001$ (LTSHAM vs. LTDMM). Statistical analysis was performed using Student's $t$ test (D, F–H), Mann–Whitney test (E, M), and one-way ANOVA test (N–S). Data are shown as mean ± SD (error bar) (*$P < 0.05$, **$P < 0.01$, ***$P < 0.001$). Source data are available online for this figure.

(Fig. 2F,H), as well as in cartilage tissue of DMM mice in the low-temperature group (Fig. 2E,G). APOE expression was also lower in OA chondrocytes and cartilage compared to controls at 33 °C (Fig. 2E–H). We also collected cartilage samples from knee osteoarthritis patients in both southern and northern regions of China who underwent total knee arthroplasty. Demographic data showed no significant differences in BMI, serum triglycerides, total cholesterol, or low-density lipoprotein levels between OA patients in the two regions (Table EV1). As seen in the temperature map of China, northern cities had lower average temperatures compared to southern cities (Fig. 2I). During December 2019 to February 2020, knee articular temperatures of OA patients were measured using a thermoprobe after 8 ~ 12 h of outdoor activity. Interestingly, the latitude-temperature trend plot revealed that ambient temperature gradually decreases with increasing latitude, while articular temperature shows a declining trend that stabilizes around 33 °C. Using 34°N as the demarcation, recorded average ambient temperatures in northern vs. southern Chinese cities were $-0.1$ °C $\pm$ 3.92 °C vs. 11.10 °C $\pm$ 4.38 °C, while articular temperatures in patients measured 33.67 °C $\pm$ 0.99 °C vs. 36.33 °C $\pm$ 0.96 °C (northern vs. southern, $P < 0.001$) respectively (Fig. 2K,L). Safranin O staining revealed more severe damage in the medial compartment of tibial plateau cartilage compared to the lateral compartment. Cartilage damage and Mankin scores were higher in the medial compartment of OA patients from northern cities than in those from southern cities (Fig. 2J,M). Additionally, COLII expression was lower and MMP13 expression was higher in the northern patients medial compartments (Fig. 2J,N,O), indicating more matrix degradation. APOE expression was found to be lower in the medial compartment and decreased in cartilage from northern cities with lower temperatures (Fig. 2J,P). This suggests that APOE expression is diminished in cartilage exhibiting greater damage and is downregulated by lower temperatures. These findings indicate that APOE expression, which was downregulated by low temperatures, may have a protective role in the progression of OA.

## Apoe knockout in cartilage exacerbates the progression of OA under cold exposure

Due to the reduced expression of APOE at low temperatures and its negative correlation with OA severity, we further investigated the role of *Apoe* in OA progression. Since global *Apoe* knockout in mice can cause hyperlipidemia, a crucial factor in OA progression, we generated *Apoe* conditional knockout *Acan*-Cre$^{ERT2}$ mice for additional study (Fig. 3A). Lipid concentrations did not differ significantly in mice exposed to varying temperatures, but *Apoe* KO mice had higher TG, TC, and LDL levels within the same temperature groups compared to other strains (Fig. 3B). In contrast, *Apoe* conditional knockout mice (*Apoe$^{-/-}$*) showed no notable differences in blood lipid levels compared to *Apoe$^{f/f}$* mice. To minimize hyperlipidemia bias in OA research, we selected *Apoe* conditional knockout mice (*Apoe$^{-/-}$*) for further analysis. PCR genotyping and IHC confirmed the efficiency of *Apoe* knockout in various tissues (Figs. 3C and EV2A–D).

As *Acan* gene may also express in microglia in dorsal root, which plays a key role in the development of OA and related pain, we firstly analyzed DRG tissues from *Apoe* conditional knockout (*Apoe$^{-/-}$*) and control (*Apoe$^{fl/fl}$* littermate) mice. Immunostaining confirmed successful *Apoe* deletion in ACAN$^+$ DRG microglia of *Apoe$^{-/-}$* mice (Fig. EV3A). To assess pain-related behavioral changes, CatWalk gait analysis revealed that DMM mice exhibited reduced paw intensity and contact area in the affected limb compared to Sham controls, consistent with OA-associated pain. However, *Apoe$^{-/-}$* mice showed no significant differences in these parameters versus *Apoe$^{f/f}$* littermates, indicating that *Apoe* deletion in ACAN$^+$ microglia does not alter pain-related behaviors (Fig. EV3B–D).

Further, our data demonstrated that DMM-induced OA mice exhibited upregulated expression of Nav1.7 (a sodium channel linked to OA progression and pain) and TAC1 (encoding substance P, a key pain-associated neurotransmitter) in the dorsal root ganglia (DRG) (Fig. EV3E–G). These findings align with established mechanisms where Nav1.7 sensitizes nociceptive neurons and substance P amplifies pain signaling in OA (Fu et al, 2024; Malek et al, 2024). However, conditional knockout of *Apoe* in ACAN$^+$ DRG microglia did not alter Nav1.7 or TAC1 expression levels, suggesting that conditional knockout of *Apoe* in these cells does not modulate these specific pain-related pathways during OA development.

Activity levels were further compared between *Apoe$^{f/f}$* and *Apoe$^{-/-}$* mice at different temperatures by open field test, revealing no significant differences except for reduced total distance traveled in low-temperature environments (Fig. EV4A–F). DMM surgery was performed on both types of mice under different temperature conditions (Fig. 3A). The body weights of the different types of

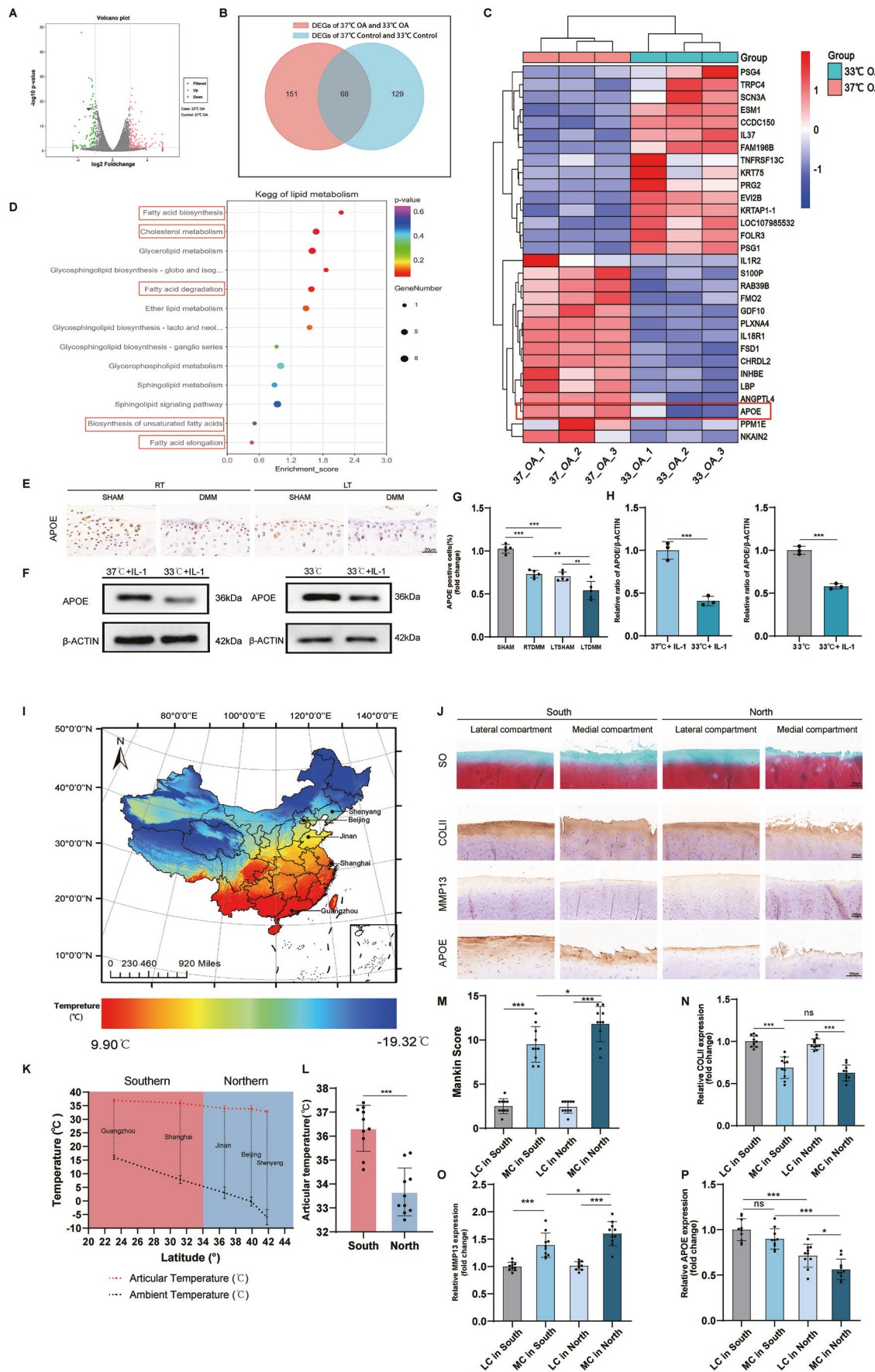

**Figure 2. *Apoe* was identified as a protective gene in OA which was downregulated by cold temperature.**

(A) The volcano plot of DEGs between 37 °C OA and 33 °C OA group ($n = 3$ patients per group). (B) The Wayne plot showed the overlapping temperature-relevant DEGs in both OA groups and non-OA groups. (C) The heatmap showed the expression level of the temperature-relevant DEGs (top 30) in 33 °C OA and 37 °C OA group. (D) KEGG analysis of DEGs related to lipid metabolism. (E) Immunohistochemistry analysis of APOE expression in the knee joint of mice. (F) Protein expression of APOE in the murine chondrocytes with IL-1β treatment and exposure of 33 °C. (G) Statistical analysis of APOE expression in mice cartilage ($n = 5$). $P < 0.0001$ (SHAM vs. RTDMM), $P < 0.0001$ (SHAM vs. LTSHAM), $P = 0.0022$ (RTDMM vs. LTDMM), $P = 0.0080$ (LTSHAM vs. LTDMM). (H) Protein analysis of APOE in chondrocytes ($n = 3$ each). $P = 0.0009$ (33 °C + IL-1 vs. 37 °C + IL-1), $P = 0.0002$ (33 °C vs. 33 °C + IL-1). (I) Map of average temperature in China from December, 2019 to February, 2020. (J) Immunohistochemistry staining of APOE, COLII, MMP13 and Safranin O/Fast Green staining of medial and lateral compartments of human knee joints from Southern or Northern China. (K) Latitude-temperature trend plot showing the relationship between knee temperature in osteoarthritis (OA) patients, ambient environmental temperature, and geographic latitude across northern and southern regions of China ($n = 4$ for Guangzhou, $n = 6$ for Shanghai, n = 4 for Jinan, $n = 4$ for Beijing and $n = 2$ for Shenyang). (L) Statistical analysis of articular temperature from Northern and Southern cities ($n = 10$ each). $P < 0.0001$. (M) Mankin score of human knee cartilage ($n = 10$). $P < 0.0001$ (LC in South vs. MC in South), $P = 0.0264$ (MC in South vs. MC in North), $P < 0.0001$ (LC in North vs. MC in North). (N) Immunohistochemistry analysis of COLII level in human cartilage ($n = 10$). $P < 0.0001$ (LC in South vs. MC in South), $P = 0.4458$ (MC in South vs. MC in North), $P < 0.0001$ (LC in North vs. MC in North). (O) Immunohistochemistry analysis of MMP13 level in human cartilage ($n = 10$). $P < 0.0001$ (LC in South vs. MC in South), $P = 0.0349$ (MC in South vs. MC in North), $P < 0.0001$ (LC in North vs. MC in North). (P) Immunohistochemistry analysis of APOE level in human cartilage ($n = 10$). $P = 0.2498$ (LC in South vs. MC in South), $P < 0.0001$ (LC in South vs. LC in North), $P < 0.0001$ (MC in South vs. MC in North), $P = 0.0356$ (LC in North vs. MC in North). Statistical analysis was performed using the Wald test within the DESeq2 framework (A), one-way ANOVA test (G, N–P), Mann–Whitney test (M), and Student's *t* test (H, L). Data are shown as mean ± SD (error bar) (*$P < 0.05$, **$P < 0.01$, ***$P < 0.001$). Source data are available online for this figure.

mice in the various temperature groups did not exhibit significant differences (Appendix Fig. S2).

In the low-temperature group, $Apoe^{-/-}$ DMM mice showed reduced safranin O staining compared to $Apoe^{f/f}$ littermates. Additionally, $Apoe^{-/-}$ DMM mice displayed more severe cartilage damage under the same temperature conditions. OARSI scores were higher in $Apoe^{-/-}$ DMM mice compared to their $Apoe^{f/f}$ counterparts (Fig. 3D). $Apoe^{-/-}$ mice exhibited lower COLII and higher MMP13 expression than $Apoe^{f/f}$ littermates at same temperatures (Fig. 3E,F). qPCR showed reduced *Col2a1* and *Acan* mRNA levels, and increased *Mmp13* and *Adamts5* expression at low temperature, further influenced by *Apoe* deficiency in cartilage (Appendix Fig. S3A–D).

In vitro studies on IL-1β-treated chondrocytes from $Apoe^{f/f}$ and $Apoe^{-/-}$ mice at 33 °C and 37 °C demonstrated decreased proliferation rates at 33 °C. OD values were lower in $Apoe^{-/-}$ chondrocytes, indicating hindered proliferation compared to $Apoe^{f/f}$ chondrocytes (Fig. 3H). Western blot analysis showed decreased COLII levels and increased MMP13 levels at 33 °C, with $Apoe^{-/-}$ chondrocytes displaying further reduction in COLII and increase in MMP13 levels at both temperatures compared to $Apoe^{f/f}$ chondrocytes (Fig. 3I,J). Alcian blue activity was also lower in the 33 °C group, particularly in IL-1β-induced $Apoe^{-/-}$ chondrocytes (Fig. 3G). These results indicated that the deficiency in the *Apoe* gene, which was downregulated in response to low temperature, exacerbated the progression of OA, confirming its protective role in OA.

## Downregulation of *Apoe* in cold exposure induced lipid accumulation, generated ROS and increased apoptosis in chondrocytes

Due to the crucial role of *Apoe* in lipid metabolism and its downregulation in chondrocytes at lower temperatures, we performed further investigation into lipid accumulation using BODIPY staining in cartilage from DMM $Apoe^{f/f}$ and $Apoe^{-/-}$ mice under varying temperature conditions. Mice with high-fat diets were included as a positive control for comparison. As illustrated in Fig. 4A,E, high-fat diet and DMM surgery led to increased lipid accumulation in cartilage, with the low temperature group

exhibiting higher levels of lipid accumulation compared to room temperature. $Apoe^{-/-}$ mice displayed significantly elevated lipid accumulation compared to $Apoe^{f/f}$ mice, indicating a potential exacerbation of lipid accumulation in cartilage under low temperatures, particularly in $Apoe^{-/-}$ mice. Further investigation using BODIPY staining in the proximal and distal tibia of mice revealed a decrease in lipid content in the proximal tibia at low temperatures for both types of DMM mice, with no significant difference in distal tibia lipid content between room and low temperature groups.

Micro-CT analysis showed a reduction in tibial trabecular bone volume following DMM surgery, without significant differences between room temperature and low temperature groups of $Apoe^{-/-}$ or $Apoe^{f/f}$ (Fig. 4B,F). TRAP staining (Fig. 4C) indicated increased $TRAP^+$ regions in the subchondral bone of the DMM surgery group compared to control groups, with no differences based on temperature or *Apoe* status (Fig. 4G). TUNEL staining revealed higher rates of apoptosis in the cartilage of DMM mice, particularly in the low temperature group and $Apoe^{-/-}$ mice (Fig. 4D,H).

Prior research has indicated that excessive cholesterol can elevate the production of reactive oxygen species (ROS) in chondrocytes, suggesting mitochondrial dysfunction and ultimately leading to chondrocyte apoptosis (Farnaghi et al, 2017). In this study, we hypothesized that the accumulation of lipids in the low-temperature group would exacerbate the severity of osteoarthritis (OA) in a similar manner. Chondrocytes cultured at 33 °C exhibited increased intracellular lipids (Fig. 4I) and ROS production (Fig. 4J) compared to 37 °C, with *Apoe* deficiency further enhancing lipid accumulation and ROS levels. Mitochondrial dysfunction was exacerbated at 33 °C (Fig. 4K), particularly in $Apoe^{-/-}$ chondrocytes, leading to increased apoptosis (Fig. 4L).

To dissect the downstream pathways of *Apoe* deficiency, we performed RNA sequencing on wild-type (WT) and *Apoe*-knockout (KO) chondrocytes, identifying 384 differentially expressed genes ($P < 0.05$ and |$\log_2$Foldchange| > 1) (Fig. 4N). Gene Set Enrichment Analysis (GSEA) highlighted significant upregulation of ROS-related pathways such as ROS production and oxidative damage response in APOE-KO cells (Fig. 4M). The ROS-related genes, such as *Cybb*, *Ncf1*, *Ncf2*, *Ncf4*, *Nos2*, *Nos3*, were markedly elevated in *Apoe* KO chondrocytes (Fig. 4O,P). Among these genes,

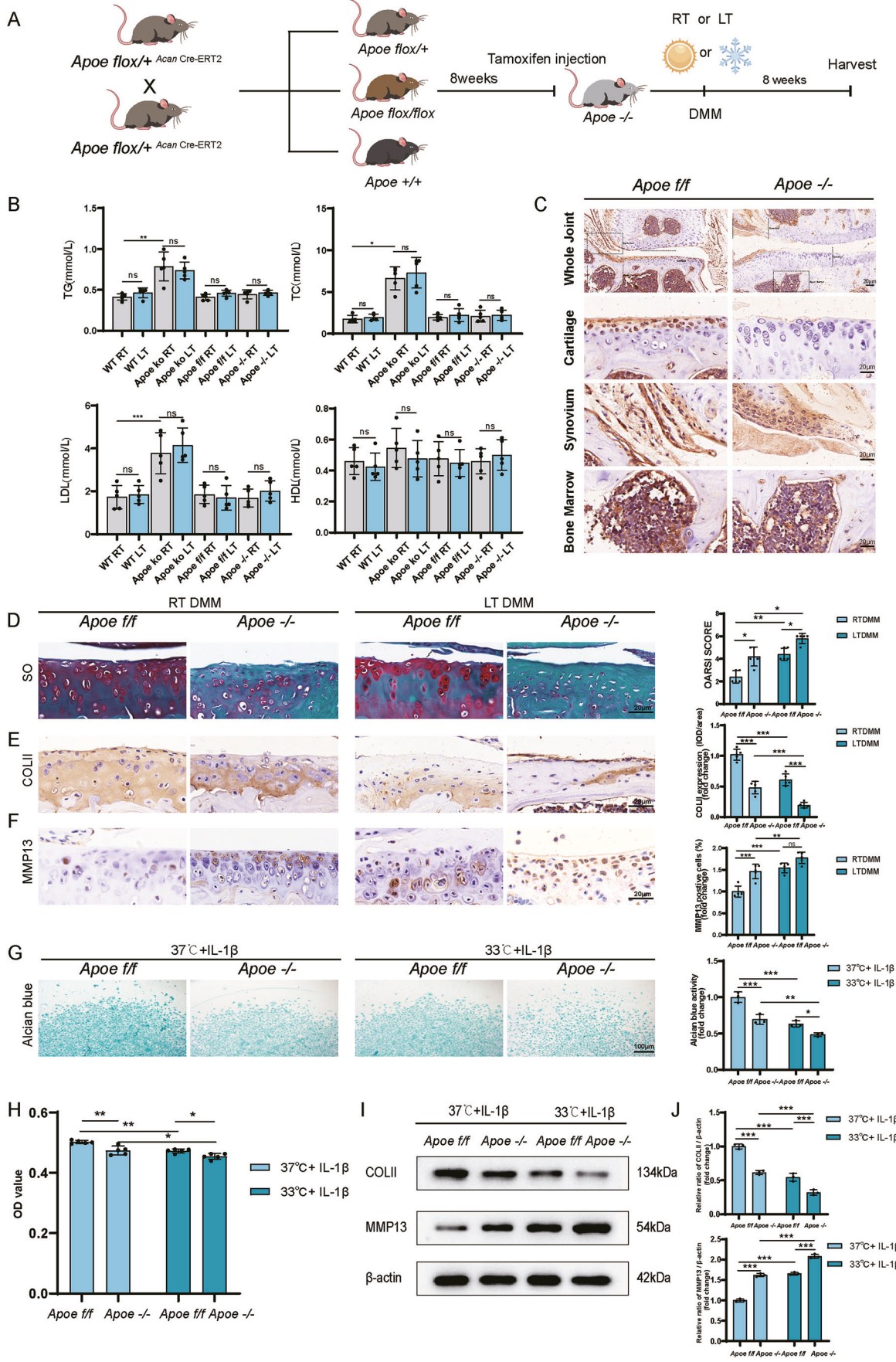

◀ **Figure 3. *Apoe* deficiency led to exacerbation of OA in vivo and in vitro.**

(A) Scheme of generation of *Apoe*$^{-/-}$ mice and following establishment of OA mice model at room and low temperature. (B) Level of TG, TC, LDL and HDL in different type of mice exposed at room or low temperature ($n = 5$). $P = 0.0090$ (TG: WT RT vs. *Apoe* ko RT), $P = 0.0270$ (TC:WT RT vs. *Apoe* ko RT), $P = 0.0002$ (LDL: WT RT vs. *Apoe* ko RT). (C) Immunohistochemistry validation of conditional knock out of *Apoe* in mice cartilage. (D) Safranin O/Fast Green staining and OARSI score of the knee joints of *Apoe*$^{f/f}$ and *Apoe*$^{-/-}$ mice at RT or LT ($n = 5$). $P = 0.0238$ (RTDMM *Apoe*$^{f/f}$ vs. RTDMM *Apoe*$^{-/-}$), $P = 0.0238$ (LTDMM *Apoe*$^{f/f}$ vs. LTDMM *Apoe*$^{-/-}$), $P = 0.0079$ (RTDMM *Apoe*$^{f/f}$ vs. LTDMM *Apoe*$^{f/f}$), $P = 0.0238$ (RTDMM *Apoe*$^{-/-}$ vs. LTDMM *Apoe*$^{-/-}$). (E) Immunohistochemistry staining and analysis of COLII level in mice cartilage ($n = 5$). $P < 0.0001$ (RTDMM *Apoe*$^{f/f}$ vs. RTDMM *Apoe*$^{-/-}$), $P < 0.0001$ (LTDMM *Apoe*$^{f/f}$ vs. LTDMM *Apoe*$^{-/-}$), $P < 0.0001$ (RTDMM *Apoe*$^{f/f}$ vs. LTDMM *Apoe*$^{f/f}$), $P = 0.0005$ (RTDMM *Apoe*$^{-/-}$ vs. LTDMM *Apoe*$^{-/-}$). (F) Immunohistochemistry staining and analysis of MMP13 level in mice cartilage ($n = 5$). $P = 0.0003$ (RTDMM *Apoe*$^{f/f}$ vs. RTDMM *Apoe*$^{-/-}$), $P = 0.0633$ (LTDMM *Apoe*$^{f/f}$ vs. LTDMM *Apoe*$^{-/-}$), $P < 0.0001$ (RTDMM *Apoe*$^{f/f}$ vs. LTDMM *Apoe*$^{f/f}$), $P = 0.0096$ (RTDMM *Apoe*$^{-/-}$ vs. LTDMM *Apoe*$^{-/-}$). (G) Alcian blue staining and analysis of chondrocytes ($n = 3$). $P = 0.0006$ (37 °C + IL-1β *Apoe*$^{f/f}$ vs. 37 °C + IL-1β *Apoe*$^{-/-}$), $P = 0.0415$ (33 °C + IL-1β *Apoe*$^{f/f}$ vs. 33 °C + IL-1β *Apoe*$^{-/-}$), $P = 0.0002$ (37 °C + IL-1β *Apoe*$^{f/f}$ vs. 33 °C + IL-1β *Apoe*$^{f/f}$), $P = 0.0064$ (37 °C + IL-1β *Apoe*$^{-/-}$ vs. 33 °C + IL-1β *Apoe*$^{-/-}$). (H) CCK8 assay of the chondrocytes ($n = 5$). $P = 0.0015$ (37 °C + IL-1β *Apoe*$^{f/f}$ vs. 37 °C + IL-1β *Apoe*$^{-/-}$), $P = 0.0392$ (33 °C + IL-1β *Apoe*$^{f/f}$ vs. 33 °C + IL-1β *Apoe*$^{-/-}$), $P = 0.0010$ (37 °C + IL-1β *Apoe*$^{f/f}$ vs. 33 °C + IL-1β *Apoe*$^{f/f}$), $P = 0.0250$ (37 °C + IL-1β *Apoe*$^{-/-}$ vs. 33 °C + IL-1β *Apoe*$^{-/-}$). (I) Result of western blot of COLII and MMP13 of the chondrocytes. (J) Protein analysis of COLII and MMP13 ($n = 3$). $P < 0.0001$ (COLII: 37 °C + IL-1β *Apoe*$^{f/f}$ vs. 37 °C + IL-1β *Apoe*$^{-/-}$), $P = 0.0009$ (COLII: 33 °C + IL-1β *Apoe*$^{f/f}$ vs. 33 °C + IL-1β *Apoe*$^{-/-}$), $P < 0.0001$ (COLII: 37 °C + IL-1β *Apoe*$^{f/f}$ vs. 33 °C + IL-1β *Apoe*$^{f/f}$), $P = 0.0001$ (COLII:37 °C + IL-1β *Apoe*$^{-/-}$ vs. 33 °C + IL-1β *Apoe*$^{-/-}$), $P < 0.0001$ (MMP13: 37 °C + IL-1β *Apoe*$^{f/f}$ vs. 37 °C + IL-1β *Apoe*$^{-/-}$), $P < 0.0001$ (MMP13: 37 °C + IL-1β *Apoe*$^{f/f}$ vs. 33 °C + IL-1β *Apoe*$^{f/f}$), $P < 0.0001$ (MMP13:37 °C + IL-1β *Apoe*$^{-/-}$ vs. 33 °C + IL-1β *Apoe*$^{-/-}$). Statistical analysis was performed using Kruskal–Wallis test (TG, TC, HDL in (B)), Mann–Whitney test (D), and one-way ANOVA test (LDL in (B, E–G, H, J)). Data are shown as mean ± SD (error bar) (*$P < 0.05$, **$P < 0.01$, ***$P < 0.001$). Source data are available online for this figure.

*Cybb* encodes NOX2, a core catalytic subunit of the NADPH oxidase (NOX) family. The NOX family represents the only enzyme class primarily dedicated to reactive oxygen species generation, playing critical roles in redox signaling and oxidative stress regulation (Vermot et al, 2021).

Collectively, these findings demonstrate that low-temperature exposure exacerbates osteoarthritis progression by downregulating APOE expression, inducing lipid accumulation in chondrocytes, which subsequently drives mitochondrial dysfunction and potentiates apoptotic pathways. Notably, this cold-induced pathological cascade is markedly amplified by *Apoe* deficiency, as APOE downregulation under cold exposure facilitates activation of ROS-generating pathways.

## RGX-104 scavenged lipid accumulation, decreased ROS production and abolished apoptosis in *Apoe*$^{+/-}$ chondrocytes at low temperature

To confirm the involvement of *Apoe* gene downregulation and lipid accumulation in OA exacerbation caused by low temperature, RGX-104, a potent LXRβ agonist, which activates the LXR/APOE signaling axis to drive transcriptional upregulation of APOE (Tavazoie et al, 2018), were incubated with chondrocytes under IL-1β stimulation at 33 °C. Due to the complete knockout of *Apoe* gene in *Apoe*$^{-/-}$ chondrocytes, activating *Apoe* with RGX-104 was not feasible. Therefore, rescue experiment used *Apoe*$^{+/-}$ instead of *Apoe*$^{-/-}$ chondrocytes. In IL-1β-induced chondrocytes cultured at 33 °C versus 37 °C, LXRβ expression remained comparable in *Apoe*$^{f/f}$ and *Apoe*$^{+/-}$ genotypes, whereas APOE expression was markedly reduced in both. Notably, under 33 °C, *Apoe*$^{+/-}$ chondrocytes displayed exacerbated degenerative phenotypes relative to *Apoe*$^{f/f}$ controls, characterized by further diminished COL II expression, heightened MMP13 expression, and amplified NOX2 expression—a key regulator of ROS production. Treatment with RGX-104 effectively increased LXRβ expression, restored APOE levels to those observed at 37 °C, rescued COL II loss, and suppressed both MMP13 and NOX2 in both genotypes under cold conditions (Fig. 5A–D). Collectively, these results demonstrate that while cold stress induces APOE suppression independent of LXRβ expression

levels, RGX-104-mediated LXRβ activation restores APOE expression, which in turn rescues chondrocyte degeneration under cold stress, as demonstrated by the recovery of COL II alongside suppression of MMP13 and NOX2. Alcian blue staining revealed reduced activity in *Apoe*$^{+/-}$ cells at 37 °C compared to *Apoe*$^{f/f}$ cells, with a reversal upon RGX-104 treatment at 33 °C (Fig. 5E,J). BODIPY staining demonstrated a significant increase in lipid accumulation in IL-1β-induced *Apoe*$^{+/-}$ chondrocytes compared to *Apoe*$^{f/f}$ chondrocytes, with higher levels observed at 33 °C than at 37 °C, mitigated by RGX-104 treatment (Fig. 5F,K). Exposure to 33 °C increased ROS levels, mitochondrial dysfunction, and apoptosis in both cell types, with *Apoe*$^{+/-}$ cells showing more severe effects (Fig. 5G–I). RGX-104 treatment alleviated these effects (Fig. 5L–N). These findings delineate a cold stress-aggravated OA pathway wherein reduced APOE expression (uncoupled from LXRβ levels) drives chondrocyte degeneration through lipid dysmetabolism, oxidative stress, and mitochondrial apoptosis. RGX-104 counteracts this cascade via LXRβ-mediated APOE restoration, coordinately rescuing extracellular matrix synthesis, suppressing catabolic and oxidative effectors, while normalizing lipid homeostasis and mitochondrial integrity, ultimately attenuating cold-exacerbated chondrocyte apoptosis.

## Cartilage degeneration at low temperature is ameliorated by RGX-104 in DMM mice

We then examined the impact of the *Apoe* agonist RGX-104 on mitigating cartilage degeneration in murine osteoarthritis models under hypothermic conditions. Using DMM mice models of *Apoe*$^{f/f}$ and *Apoe*$^{+/-}$, we administered RGX-104 intra-articularly at 4 weeks post-surgery for 4 weeks (Fig. 6A). APOE expression in cartilage was significantly decreased in *Apoe*$^{+/-}$ mice (Fig. 6B), confirming the successful generation of the heterozygous model. In the LT DMM mouse model, cartilage LXRβ expression showed no significant difference compared to the RT DMM group. However, intra-articular injection of RGX-104 elevated LXRβ expression in both *Apoe*$^{f/f}$ and *Apoe*$^{+/-}$ groups (Fig. 6C). APOE levels were lower in DMM mice at room temperature but further decreased at low temperature, restored by RGX-104 treatment (Fig. 6D). These

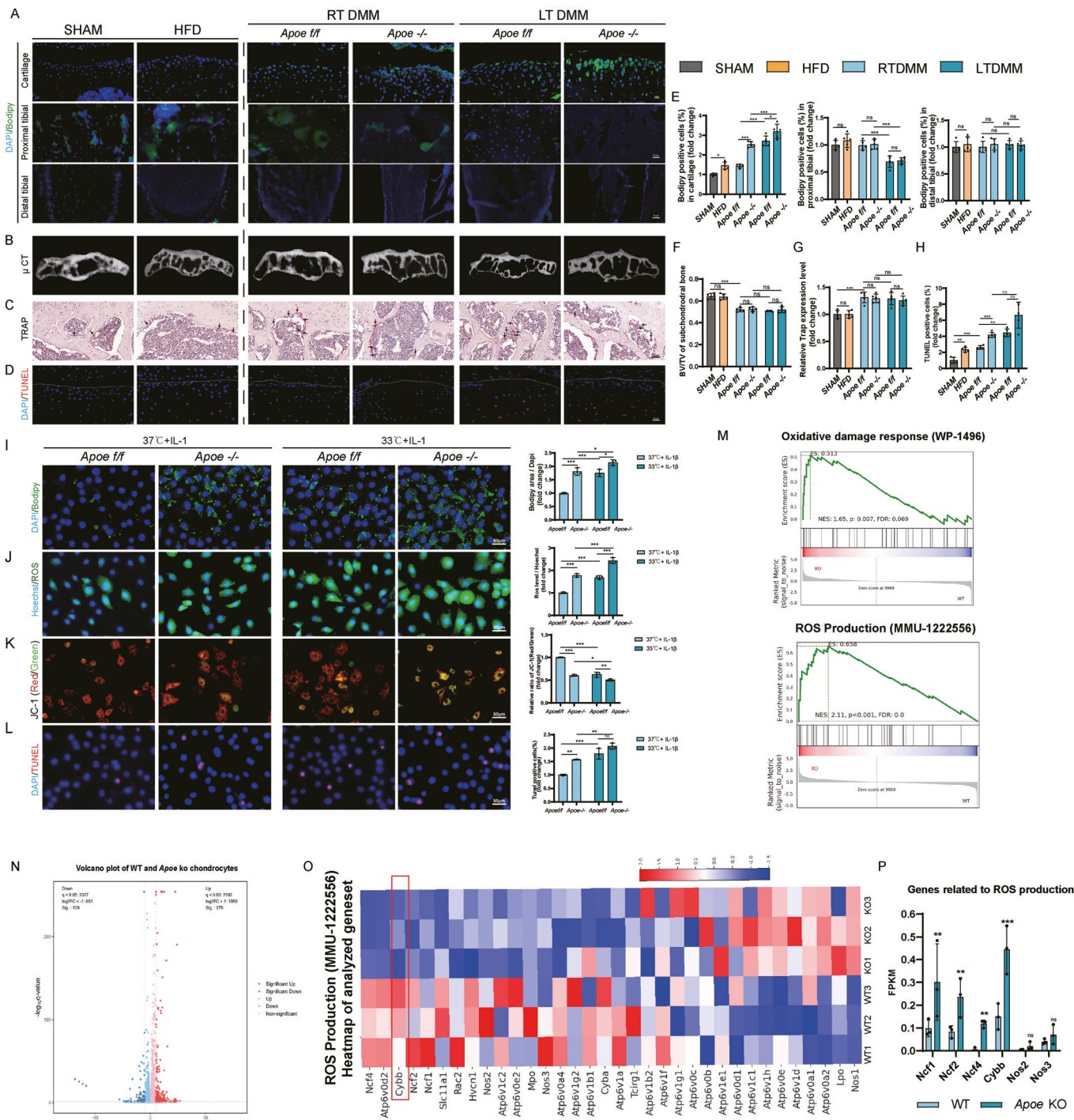

results suggest that cold stress downregulates APOE independently of LXRβ expression, yet LXRβ activation via RGX-104 remains a potent compensatory strategy to counteract cold-induced APOE suppression. RGX-104 injection protected cartilage structure and matrix content, resulting in lower OARSI scores in both *Apoe^f/f* and *Apoe^+/−* mice under low temperature (Fig. 6E). COLII levels decreased in low temperature groups, particularly in *Apoe^+/−* mice, while MMP13 levels increased, especially in *Apoe^+/−* mice. RGX-104 increased COLII levels and reduced MMP13 levels in LT DMM

mice, indicating a protective effect on cartilage matrix maintenance (Fig. 6F,G). RGX-104 mitigated matrix degradation by upregulating *Col2a1*, *Acan*, and *Apoe*, while downregulating *Mmp13* and *Adamts5* in both *Apoe^f/f* and *Apoe^+/−* mice (Appendix Fig. S4A–E). Consistent with cellular findings, animal experiments demonstrated significantly increased NOX2 expression in the cartilage of LT-treated DMM mice of both genotype compared to the RT group, while RGX-104 intervention effectively reduced NOX2 levels in LT DMM mice (Fig. 6H). BODIPY staining showed increased lipid

◀ **Figure 4. Downregulation of *Apoe* in cold exposure induced lipid accumulation, generated ROS and increased apoptosis in chondrocytes.**

(A) BODIPY staining of cartilage, proximal and distal tibial of different types of DMM mice housed at RT or LT. (B) µCT of subchondral cartilage in mice. (C) TRAP staining of subchondral cartilage in mice. (D) TUNEL staining of cartilage in mice. (E) Statistical analysis of BODIPY staining in mice ($n = 5$). $P = 0.0216$ (Cartilage: Sham vs. HFD), $P < 0.0001$ (Cartilage: RTDMM $Apoe^{f/f}$ vs. RTDMM $Apoe^{-/-}$), $P = 0.0130$ (Cartilage: LTDMM $Apoe^{f/f}$ vs. LTDMM $Apoe^{-/-}$), $P < 0.0001$ (Cartilage: RTDMM $Apoe^{f/f}$ vs. LTDMM $Apoe^{f/f}$), $P = 0.0005$ (Cartilage: RTDMM $Apoe^{-/-}$ vs. LTDMM $Apoe^{-/-}$), $P = 0.0005$ (Proximal tibial: RTDMM $Apoe^{f/f}$ vs. LTDMM $Apoe^{f/f}$), $P = 0.0005$ (Proximal tibial: RTDMM $Apoe^{-/-}$ vs. LTDMM $Apoe^{-/-}$). (F) BV/TV analysis of subchondral cartilage in mice ($n = 5$). $P < 0.0001$ (SHAM vs. RTDMM $Apoe^{f/f}$). (G) Analysis of TRAP level in subchondral cartilage of mice ($n = 5$). $P = 0.0001$ (SHAM vs. RTDMM $Apoe^{f/f}$). (H) Analysis of TUNEL expression in cartilage of mice ($n = 5$). $P = 0.0023$ (Sham vs. HFD), $P = 0.0008$ (SHAM vs. RTDMM $Apoe^{f/f}$), $P = 0.0004$ (RTDMM $Apoe^{f/f}$ vs. RTDMM $Apoe^{-/-}$), $P = 0.1892$ (LTDMM $Apoe^{f/f}$ vs. LTDMM $Apoe^{-/-}$), $P = 0.0021$ (RTDMM $Apoe^{f/f}$ vs. LTDMM $Apoe^{f/f}$), $P = 0.1364$ (RTDMM $Apoe^{-/-}$ vs. LTDMM $Apoe^{-/-}$). (I) BODIPY staining and analysis of chondrocytes ($n = 3$). $P < 0.0001$ (37 °C + IL-1β $Apoe^{f/f}$ vs. 37 °C + IL-1β $Apoe^{-/-}$), $P = 0.0104$ (33 °C + IL-1β $Apoe^{f/f}$ vs. 33 °C + IL-1β $Apoe^{-/-}$), $P = 0.0001$ (37 °C + IL-1β $Apoe^{f/f}$ vs. 33 °C + IL-1β $Apoe^{f/f}$), $P = 0.0239$ (37 °C + IL⁻1β $Apoe^{-/-}$ vs. 33 °C + IL-1β $Apoe^{-/-}$). (J) ROS detection and analysis of chondrocytes ($n = 3$). $P < 0.0001$ (37 °C + IL-1β $Apoe^{f/f}$ vs. 37 °C + IL-1β $Apoe^{-/-}$), $P < 0.0001$ (33 °C + IL-1β $Apoe^{f/f}$ vs. 33 °C + IL-1β $Apoe^{-/-}$), $P < 0.0001$ (37 °C + IL-1β $Apoe^{f/f}$ vs. 33 °C + IL-1β $Apoe^{f/f}$), $P < 0.0001$ (37 °C + IL-1β $Apoe^{-/-}$ vs. 33 °C + IL-1β $Apoe^{-/-}$). (K) JC-1 staining and analysis of chondrocytes ($n = 3$). $P < 0.0001$ (37 °C + IL-1β $Apoe^{f/f}$ vs. 37 °C + IL-1β $Apoe^{-/-}$), $P = 0.0067$ (33 °C + IL-1β $Apoe^{f/f}$ vs. 33 °C + IL-1β $Apoe^{-/-}$), $P < 0.0001$ (37 °C + IL-1β $Apoe^{f/f}$ vs. 33 °C + IL-1β $Apoe^{f/f}$), $P = 0.0149$ (37 °C + IL-1β $Apoe^{-/-}$ vs. 33 °C + IL-1β $Apoe^{-/-}$). (L) TUNEL staining and analysis of chondrocytes ($n = 3$). $P = 0.0015$ (37 °C + IL-1β $Apoe^{f/f}$ vs. 37 °C + IL-1β $Apoe^{-/-}$), $P = 0.0690$ (33 °C + IL-1β $Apoe^{f/f}$ vs. 33 °C + IL-1β $Apoe^{-/-}$), $P <= 0.0002$ (37 °C + IL-1β $Apoe^{f/f}$ vs. 33 °C + IL-1β $Apoe^{f/f}$), $P = 0.0036$ (37 °C + IL-1β $Apoe^{-/-}$ vs. 33 °C + IL-1β $Apoe^{-/-}$). (M) Gene Set Enrichment Analysis (GSEA) results comparing wild-type (WT) and *Apoe* knockout (*Apoe* KO) chondrocytes, focusing on reactive oxygen species (ROS)-related pathways. (N) Heatmap illustrating differentially expressed genes (DEGs) between wild-type (WT) and *Apoe* knockout (*Apoe* KO) chondrocytes. (O) Heatmap depicting differentially expressed genes associated with ROS generation. (P) Bar plot showing expression levels of key differentially upregulated genes associated with ROS production in chondrocytes. $P = 0.0056$ (*Ncf1*: WT vs. *Apoe* KO), $P = 0.0045$ (*Ncf2*: WT vs. *Apoe* KO), $P = 0.0079$ (*Ncf4*: WT vs. *Apoe* KO), $P < 0.0001$ (*Cybb*: WT vs. *Apoe* KO), $P = 0.1429$ (*Nos2*: WT vs. *Apoe* KO), $P = 0.2104$ (*Nos3*: WT vs. *Apoe* KO). Statistical analysis was performed using one-way ANOVA test (E–G, I–L), Welch's ANOVA test (H). Data are shown as mean ± SD (error bar) (*$P < 0.05$, **$P < 0.01$, ***$P < 0.001$). Source data are available online for this figure.

content in cartilage under low temperatures, especially in $Apoe^{+/-}$ mice, but RGX-104 reduced lipid accumulation (Fig. 6I). TUNEL staining revealed fewer TUNEL⁺ cells in cartilage of both $Apoe^{f/f}$ and $Apoe^{+/-}$ mice under low temperature post-RGX-104 injection (Fig. 6J).

Our findings demonstrate that cold exposure exacerbates OA progression through APOE downregulation, a process independent of LXRβ expression changes, which promotes lipid dysregulation and oxidative production in chondrocytes. Pharmacological activation of LXRβ by RGX-104 effectively counteracts this pathology by restoring APOE transcription, thereby orchestrating multi-faceted cartilage protection in animal models. In DMM mice subjected to cold exposure, RGX-104 treatment not only rescued APOE levels but also attenuated progression of OA by preserving cartilage and matrix integrity, reducing lipid overload, and suppressing ROS-associated damage. This comprehensive rescue effect highlights the therapeutic potential of targeting the LXRβ-APOE axis to disrupt cold-aggravated OA pathogenesis.

## Discussion

In this investigation, we proved that the severity of OA was aggravated by low temperature through downregulation of APOE expression and an increase in lipid accumulation in OA chondrocytes, which was validated by the conditional *Apoe* knock-out mouse model. We also identified the protective effect of *Apoe* induction by RGX-104 (LXRβ agonist) on alleviating the severity of OA caused by low temperature.

Primary OA is caused by multiple risk factors, including increasing age, obesity, and articular biomechanics, with obesity and advancing age being the most notable (Martel-Pelletier et al, 2016). In this study, low temperature was found to exacerbate the progression of OA. In a previous multicenter study, a large number of OA patients reported that rainy and cold weather affected their pain (Timmermans et al, 2014). One possible explanation may be that the viscosity of synovial fluid is increased in cold temperatures,

which may contribute to the stiffness of the joint, which leads to the sensitivity of pain. Nevertheless, no research has established a particular molecular mechanism for the impact of low temperature on OA cartilage in vivo or in vitro. As far as we are aware, this is the first study to show the aggravating role of low temperature in OA through the downregulation of APOE expression that leads to increased lipid accumulation in cartilage.

In this study, we discovered that APOE downregulation at low temperatures led to increased lipid accumulation, which accelerated the progression of OA. According to earlier research, lipid metabolism is crucial for preserving the homeostasis of cartilage. According to epidemiological research, obese people have a noticeably higher incidence of OA (Song et al, 2022; Yusuf et al, 2010). Patients with OA have been found to have higher levels of free fatty acids (FFAs) (Bonner et al, 1975; Louati et al, 2015; Martel-Pelletier et al, 2016; Yusuf et al, 2010), which contributes to lipid accumulation in articular cartilage (Lippiello et al, 1991). Saturated fatty acids (SFAs) were found to contribute to OA by inducing inflammation and activating abnormal autophagy in cartilage (Rocha et al, 2017). Farnaghi et al, demonstrated that excessive cholesterol led to the overproduction of ROS, leading to mitochondrial dysfunction in chondrocytes (Farnaghi et al, 2017). Moreover, Sujeong Park et al found that PPARα is responsible for lipid accumulation in OA chondrocytes by modulating ACOT12, which is involved in de novo lipogenesis, leading to lipid accumulation and apoptosis of chondrocytes (Park et al, 2022). Choi et al reported similar results regarding cholesterol and OA, demonstrating that the CH25H-CYP7B1-RORα axis was the mechanism by which elevated cholesterol levels in chondrocytes led to OA in mice (Choi et al, 2019). Consistent with our study, these results showed that lipid accumulation occurs in OA and contributes significantly to the OA progression.

The primary gene downregulated by low temperature in the current investigation was found to be *Apoe*, which exacerbated the severity of OA. *Apoe*, one of the LXR target genes, was discovered to be participating in the process of lipid metabolism including lipid efflux and reverse cholesterol transports (Edwards et al, 2002;

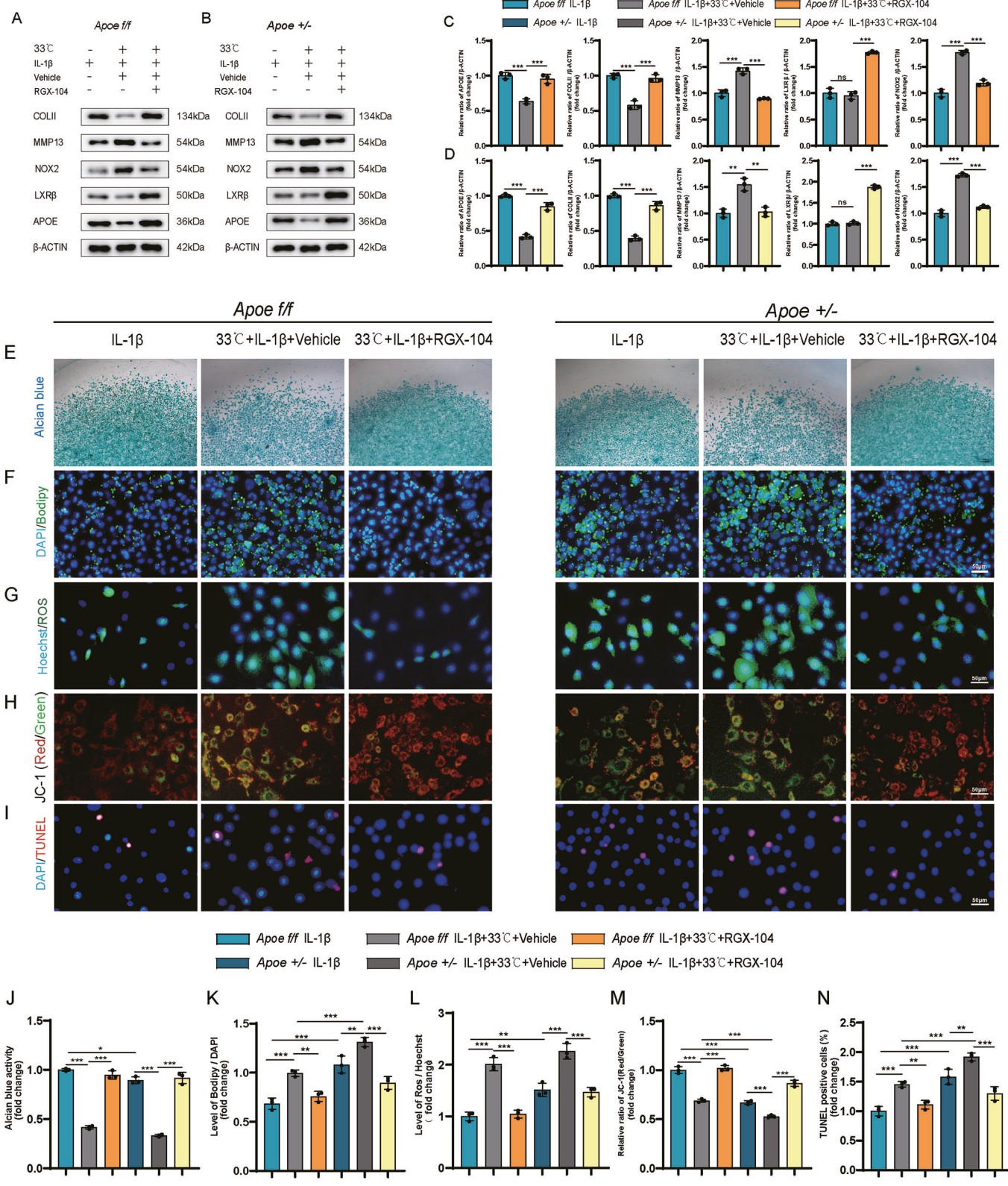

**Figure 5. The rescue study of the effect of RGX-104 on chondrocytes at low temperature.**

(A) The western blot result of COLII, MMP13, APOE, LXRβ, NOX2 in *Apoe*$^{f/f}$ chondrocytes. (B) The western blot result of COLII, MMP13, APOE, LXRβ, NOX2 in *Apoe*$^{+/-}$ chondrocytes. (C) The analysis of protein expression in *Apoe*$^{f/f}$ chondrocytes ($n = 3$). $P = 0.0003$ (APOE: IL-1β vs. IL-β + 33 °C+Vehicle), $P = 0.0005$ (APOE: Vehicle vs. RGX-104), $P < 0.0001$ (COLII: IL-1β vs. IL-β + 33 °C+Vehicle), $P < 0.0001$ (COLII: Vehicle vs. RGX-104), $P = 0.0004$ (MMP13: IL-1β vs. IL-β + 33 °C+Vehicle), $P = 0.0001$ (MMP13: Vehicle vs. RGX-104), $P = 0.7045$ (LXRβ: IL-1β vs. IL-β + 33 °C+Vehicle), $P < 0.0001$ (LXRβ: Vehicle vs. RGX-104), $P < 0.0001$ (NOX2: IL-1β vs. IL-β + 33 °C +Vehicle), $P < 0.0001$ (NOX2: Vehicle vs. RGX-104). (D) The analysis of protein expression in *Apoe*$^{+/-}$ chondrocytes ($n = 3$). $P < 0.0001$ (APOE: IL-1β vs. IL-β + 33 °C +Vehicle), $P < 0.0001$ (APOE: Vehicle vs. RGX-104), $P < 0.0001$ (COLII: IL-1β vs. IL-β + 33 °C+Vehicle), $P < 0.0001$ (COLII: Vehicle vs. RGX-104), $P = 0.0011$ (MMP13: IL-1β vs. IL-β + 33 °C+Vehicle), $P = 0.0014$ (MMP13: Vehicle vs. RGX-104), $P = 0.9075$ (LXRβ: IL-1β vs. IL-β + 33 °C+Vehicle), $P < 0.0001$ (LXRβ: Vehicle vs. RGX-104), $P < 0.0001$ (NOX2: IL-1β vs. IL-β + 33 °C+Vehicle), $P < 0.0001$ (NOX2: Vehicle vs. RGX-104). (E) Alcian blue staining of chondrocytes. (F) BODIPY staining of chondrocytes. (G) Detection of ROS production in chondrocytes. (H) JC-1 staining of chondrocytes. (I) TUNEL staining of chondrocytes ($n = 3$). (J) Analysis of alcian blue staining ($n = 3$). $P < 0.0001$ (*Apoe*$^{f/f}$ IL-1β vs. *Apoe*$^{f/f}$ IL-1β + 33 °C+Vehicle), $P < 0.0001$ (*Apoe*$^{f/f}$ Vehicle vs. *Apoe*$^{f/f}$ RGX-104), $P < 0.0001$ (*Apoe*$^{+/-}$ IL-1β vs. *Apoe*$^{+/-}$ IL-1β + 33 °C+Vehicle), $P < 0.0001$ (*Apoe*$^{+/-}$ Vehicle vs. *Apoe*$^{+/-}$ RGX-104), $P = 0.0210$ (*Apoe*$^{f/f}$ IL-1β vs. *Apoe*$^{+/-}$ IL-1β). (K) Analysis of BODIPY staining ($n = 3$). $P = 0.0003$ (*Apoe*$^{f/f}$ IL-1β vs. *Apoe*$^{f/f}$ IL-1β + 33 °C+Vehicle), $P = 0.0037$ (*Apoe*$^{f/f}$ Vehicle vs. *Apoe*$^{f/f}$ RGX-104), $P = 0.0046$ (*Apoe*$^{+/-}$ IL-1β vs. *Apoe*$^{+/-}$ IL-1β + 33 °C+Vehicle), $P < 0.0001$ (*Apoe*$^{+/-}$ Vehicle vs. *Apoe*$^{+/-}$ RGX-104), $P < 0.0001$ (*Apoe*$^{f/f}$ IL-1β vs. *Apoe*$^{+/-}$ IL-1β), $P = 0.0003$ (*Apoe*$^{f/f}$ Vehicle vs. *Apoe*$^{+/-}$ Vehicle). (L) Analysis of ROS in chondrocytes ($n = 3$). $P < 0.0001$ (*Apoe*$^{f/f}$ IL-1β vs. *Apoe*$^{f/f}$ IL-1β + 33 °C+Vehicle), $P < 0.0001$ (*Apoe*$^{f/f}$ Vehicle vs. *Apoe*$^{f/f}$ RGX-104), $P < 0.0001$ (*Apoe*$^{+/-}$ IL-1β vs. *Apoe*$^{+/-}$ IL-1β + 33 °C+Vehicle), $P < 0.0001$ (*Apoe*$^{+/-}$ Vehicle vs. *Apoe*$^{+/-}$ RGX-104), $P = 0.0014$ (*Apoe*$^{f/f}$ IL-1β vs. *Apoe*$^{+/-}$ IL-1β). (M) Analysis of JC-1 staining ($n = 3$). $P < 0.0001$ (*Apoe*$^{f/f}$ IL-1β vs. *Apoe*$^{f/f}$ IL-1β + 33 °C+Vehicle), $P < 0.0001$ (*Apoe*$^{f/f}$ Vehicle vs. *Apoe*$^{f/f}$ RGX-104), $P = 0.0002$ (*Apoe*$^{+/-}$ IL-1β vs. *Apoe*$^{+/-}$ IL-1β + 33 °C+Vehicle), $P < 0.0001$ (*Apoe*$^{+/-}$ Vehicle vs. *Apoe*$^{+/-}$ RGX-104), $P < 0.0001$ (*Apoe*$^{f/f}$ IL-1β vs. *Apoe*$^{+/-}$ IL-1β), $P < 0.0001$ (*Apoe*$^{f/f}$ Vehicle vs. *Apoe*$^{+/-}$ Vehicle). (N) Analysis of TUNEL staining ($n = 3$). $P = 0.0005$ (*Apoe*$^{f/f}$ IL-1β vs. *Apoe*$^{f/f}$ IL-1β + 33 °C+Vehicle), $P = 0.0053$ (*Apoe*$^{f/f}$ Vehicle vs. *Apoe*$^{f/f}$ RGX-104), $P = 0.0066$ (*Apoe*$^{+/-}$ IL-1βvs. *Apoe*$^{+/-}$ IL-1β + 33 °C+Vehicle), $P < 0.0001$ (*Apoe*$^{+/-}$ Vehicle vs. *Apoe*$^{+/-}$ RGX-104), $P < 0.0001$ (*Apoe*$^{f/f}$ IL-1β vs. *Apoe*$^{+/-}$ IL-1β), $P = 0.0004$ (*Apoe*$^{f/f}$ Vehicle vs. *Apoe*$^{+/-}$ Vehicle). Statistical analysis was performed using one-way ANOVA test (C, D, J–N). Data are shown as mean ± SD (error bar) (*$P < 0.05$, **$P < 0.01$,***$P < 0.001$). Source data are available online for this figure.

Farnaghi et al, 2017; Ouimet et al, 2019; Piedrahita et al, 1992; Sankaranarayanan et al, 2009; Vedhachalam et al, 2007). *Apoe* knock out mouse has long been used as a powerful tool to induce the model of cardiovascular diseases for its essential role in lipid elimination (Piedrahita et al, 1992). Interestingly, previous studies have shown that APOE was downregulated in OA cartilage and *Apoe* deficiency was connected to the development of OA (Archer et al, 2016; Collins-Racie et al, 2009; de Munter et al, 2016; Farnaghi et al, 2017), indicating the protective role of *Apoe* in OA. However, no study has demonstrated the effect of low temperature on APOE level and whether *Apoe* deficiency would accelerate the progression of OA by changing cartilage lipid metabolism. In this study, we first found that Low temperature exacerbates OA via *Apoe* down-regulation and lipid accumulation. Liver X receptors (LXRβ and LXRα), members of the nuclear hormone receptor family, are key transcriptional activators of *Apoe* and other lipid metabolism-related genes, primarily regulating lipid efflux (Das et al, 2025; Tavazoie et al, 2018). Previous studies have shown that pharmacological activation of ubiquitously expressed LXRβ significantly upregulates APOE expression and suppresses melanoma tumor progression (Bensinger et al, 2008; Fontaine et al, 2007; Zelcer et al, 2007). To investigate whether cold stress aggravated OA via APOE or LXRβ, we conducted in vitro and in vivo rescue studies via the LXRβ agonist RGX-104. Interestingly, chondrocytes LXRβ expression remained unaffected by low temperature, indicating that temperature-dependent APOE suppression operates independently of LXRβ regulation. Nevertheless, RGX-104 serves as a compensatory therapeutic strategy by reactivating cold-diminished APOE expression in chondrocytes, thereby mitigating cold-aggravated osteoarthritis pathology. RGX-104-mediated LXRβ activation restored low temperature-suppressed *Apoe* expression in both OA chondrocytes and cartilage. This restoration effectively reduced pathological lipid accumulation in chondrocytes and cartilage under cold stress, confirming the pivotal role of APOE in lipid clearance. Consequently, RGX-104 treatment alleviated cold-exacerbated OA progression by rebalancing cartilage metabolism-enhancing anabolic markers (e.g., COLII) while suppressing

catabolic factors (MMP13, ADAMTS5). Our findings position lipid overload as the central mechanism linking APOE deficiency to temperature-sensitive OA progression, with pharmacological LXRβ activation offering therapeutic intervention by restoring lipid homeostasis. Our downstream RNA-sequencing revealed that *Apoe* deficiency in chondrocytes elevated ROS production and oxidative stress pathways, marked by upregulation of Cybb (encoding NOX2), a key NADPH oxidase in ROS generation (Vermot et al, 2021). NOX2 is minimally expressed in healthy chondrocytes but is significantly upregulated in OA cartilage and IL-1β-stimulated chondrocytes (Waddington et al, 2015). Genetic ablation of NOX2 attenuates OA progression in preclinical models, reducing cartilage degeneration and synovial hyperplasia, while pharmacological inhibition of its p47phox subunit suppresses ROS overproduction and ameliorates OA pathology (Waddington et al, 2015; Nogueira-Recalde et al, 2019). Our findings align with this ROS-dependent axis, demonstrating that APOE loss exacerbates NOX2-driven oxidative stress under cold exposure. Critically, RGX-104-mediated suppression of NOX2 in cellular and animal models attenuated this pathway, highlighting its therapeutic relevance.

Numerous studies have demonstrated the protective effect of lipid-lowering drugs and LXR agonists in OA animal models. When mice with the Insig1 and Insig2 genes were given statins to inhibit intracellular cholesterol, the development of the OA phenotype and excessive cholesterol production in chondrocytes were lessened (Sekar et al, 2017). Moreover, the use of atorvastatin and mitochondrial antioxidants reversed the elevated cholesterol-related OA modifications in DMM mice (Farnaghi et al, 2017). Consistent with the results of the present study, increased LXR activation by an LXR agonist in human articular explants in vivo led to the elevated level of LXR target genes, including *Apoe*, and reduced cytokine-induced proteoglycan degradation. Moreover, Ratneswaran et al demonstrated that the LXR agonist GW3965 activated the LXR signaling pathway and decreased the expression of MMP-2, MMP-13, and ADAMTS-4, three catabolic proteases in OA chondrocytes (Ratneswaran et al, 2017). RGX-104 functions as a potent LXRβ agonist that activates the LXR/ApoE axis, driving

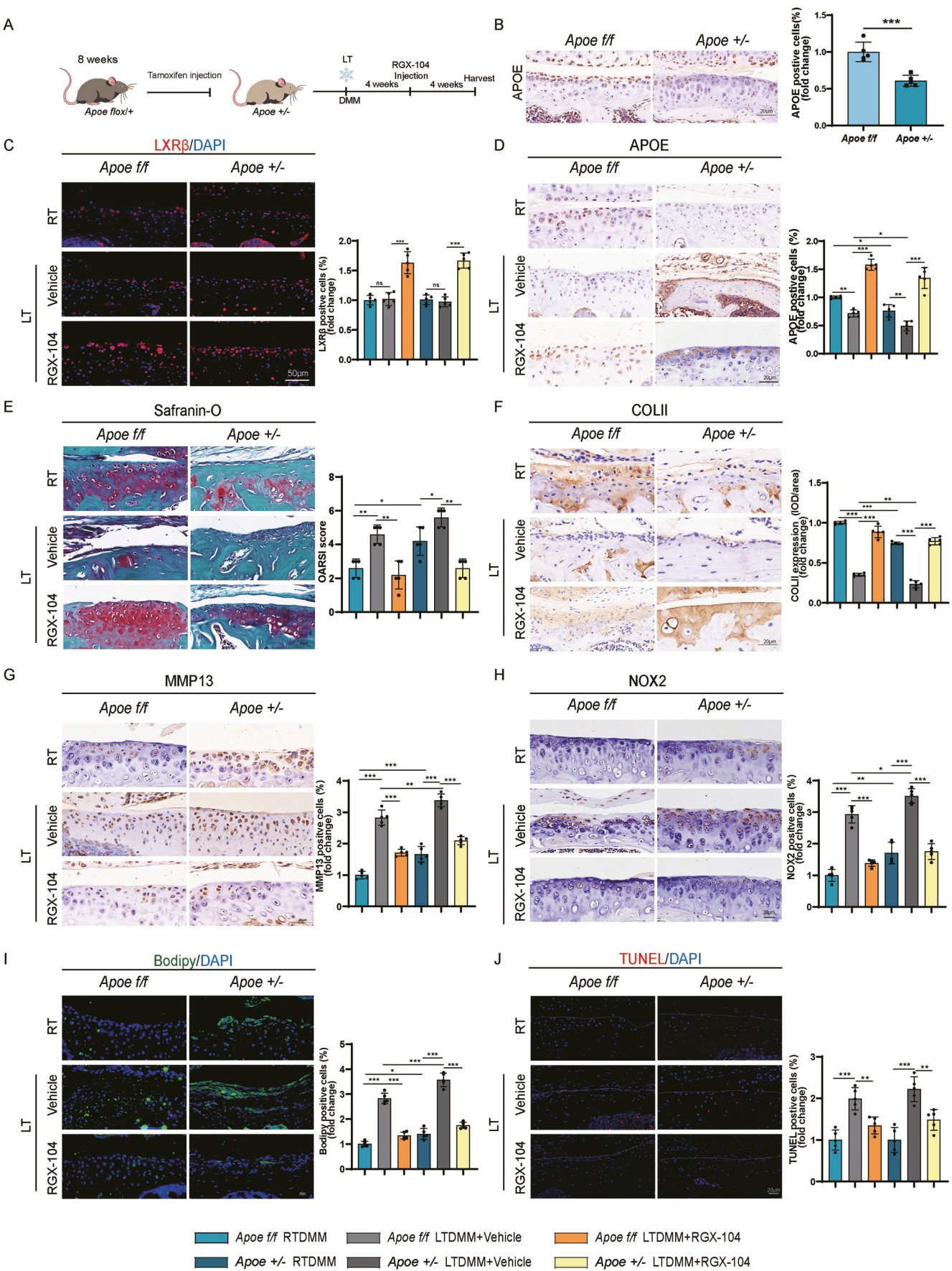

**Figure 6. The rescue study of the effect of RGX-104 injection on DMM $Apoe^{+/-}$ mice.**

(A) Scheme of generation of $Apoe^{+/-}$ mice and experimental design of RGX-104 articular injection to OA mice models at low temperature. (B) Validation of APOE expression in cartilage of $Apoe^{+/-}$ mice by IHC ($n = 5$). $P = 0.0004$. (C) LXRβ staining in cartilage and statistical analysis ($n = 5$). $P = 0.9997$ ($Apoe^{f/f}$ RT vs. $Apoe^{f/f}$ LT +Vehicle), $P < 0.0001$ ($Apoe^{f/f}$ Vehicle vs. $Apoe^{f/f}$ RGX-104), $P = 0.9965$ ($Apoe^{+/-}$ RT vs. $Apoe^{+/-}$ LT+Vehicle), $P < 0.0001$ ($Apoe^{+/-}$ Vehicle vs. $Apoe^{+/-}$ RGX-104). (D) Immunohistochemistry staining of APOE in cartilage and statistical analysis ($n = 5$). $P = 0.0040$ ($Apoe^{f/f}$ RT vs. $Apoe^{f/f}$ LT+Vehicle), $P < 0.0001$ ($Apoe^{f/f}$ Vehicle vs. $Apoe^{f/f}$ RGX-104), $P = 0.0076$ ($Apoe^{+/-}$ RT vs. $Apoe^{+/-}$ LT+Vehicle), $P < 0.0001$ ($Apoe^{+/-}$ Vehicle vs. $Apoe^{+/-}$ RGX-104), $P = 0.0179$ ($Apoe^{f/f}$ RT vs. $Apoe^{+/-}$ RT), $P = 0.0324$ ($Apoe^{f/f}$ Vehicle vs. $Apoe^{+/-}$ Vehicle). (E) Safranin O/Fast Green staining of the mice knee joints and OARSI score ($n = 5$). $P = 0.0079$ ($Apoe^{f/f}$ RT vs. $Apoe^{f/f}$ LT+Vehicle), $P = 0.0079$ ($Apoe^{f/f}$ Vehicle vs. $Apoe^{f/f}$ RGX-104), $P = 0.0476$ ($Apoe^{+/-}$ RT vs. $Apoe^{+/-}$ LT+Vehicle), $P = 0.0079$ ($Apoe^{+/-}$ Vehicle vs. $Apoe^{+/-}$ RGX-104), $P = 0.0317$ ($Apoe^{f/f}$ RT vs. $Apoe^{+/-}$ RT). (F) Immunohistochemistry staining of COLII in cartilage and statistical analysis ($n = 5$). $P < 0.0001$ ($Apoe^{f/f}$ RT vs. $Apoe^{f/f}$ LT+Vehicle), $P < 0.0001$ ($Apoe^{f/f}$ Vehicle vs. $Apoe^{f/f}$ RGX-104), $P < 0.0001$ ($Apoe^{+/-}$ RT vs. $Apoe^{+/-}$ LT+Vehicle), $P < 0.0001$ ($Apoe^{+/-}$ Vehicle vs. $Apoe^{+/-}$ RGX-104), $P < 0.0001$ ($Apoe^{f/f}$ RT vs. $Apoe^{+/-}$ RT), $P = 0.0017$ ($Apoe^{f/f}$ Vehicle vs. $Apoe^{+/-}$ Vehicle). (G) Immunohistochemistry staining of MMP13 in cartilage and statistical analysis ($n = 5$). $P < 0.0001$ ($Apoe^{f/f}$ RT vs. $Apoe^{f/f}$ LT+Vehicle), $P < 0.0001$ ($Apoe^{f/f}$ Vehicle vs. $Apoe^{f/f}$ RGX-104), $P < 0.0001$ ($Apoe^{+/-}$ RT vs. $Apoe^{+/-}$ LT+Vehicle), $P < 0.0001$ ($Apoe^{+/-}$ Vehicle vs. $Apoe^{+/-}$ RGX-104), $P = {<}0.0001$ ($Apoe^{f/f}$ RT vs. $Apoe^{+/-}$ RT), $P = 0.0011$($Apoe^{f/f}$ Vehicle vs. $Apoe^{+/-}$ Vehicle). (H) Immunohistochemistry staining of NOX2 in cartilage and statistical analysis ($n = 5$). $P < 0.0001$ ($Apoe^{f/f}$ RT vs. $Apoe^{f/f}$ LT+Vehicle), $P < 0.0001$ ($Apoe^{f/f}$ Vehicle vs. $Apoe^{f/f}$ RGX-104), $P < 0.0001$ ($Apoe^{+/-}$ RT vs. $Apoe^{+/-}$ LT +Vehicle), $P < 0.0001$ ($Apoe^{+/-}$ Vehicle vs. $Apoe^{+/-}$ RGX-104), $P = 0.0014$ ($Apoe^{f/f}$ RT vs. $Apoe^{+/-}$ RT), $P = 0.0105$ ($Apoe^{f/f}$ Vehicle vs. $Apoe^{+/-}$ Vehicle). (I) BODIPY staining in cartilage and statistical analysis ($n = 5$). $P < 0.0001$ ($Apoe^{f/f}$ RT vs. $Apoe^{f/f}$ LT+Vehicle), $P < 0.0001$ ($Apoe^{f/f}$ Vehicle vs. $Apoe^{f/f}$ RGX-104), $P < 0.0001$ ($Apoe^{+/-}$ RT vs. $Apoe^{+/-}$ LT+Vehicle), $P < 0.0001$ ($Apoe^{+/-}$ Vehicle vs. $Apoe^{+/-}$ RGX-104), $P = 0.0263$ ($Apoe^{f/f}$ RT vs. $Apoe^{+/-}$ RT), $P < 0.0001$ ($Apoe^{f/f}$ Vehicle vs. $Apoe^{+/-}$ Vehicle). (J) TUNEL staining in cartilage and statistical analysis ($n = 5$). $P < 0.0001$ ($Apoe^{f/f}$ RT vs. $Apoe^{f/f}$ LT+Vehicle), $P = 0.0083$ ($Apoe^{f/f}$ Vehicle vs. $Apoe^{f/f}$ RGX-104), $P < 0.0001$ ($Apoe^{+/-}$ RT vs. $Apoe^{+/-}$ LT+Vehicle), $P = 0.0021$ ($Apoe^{+/-}$ Vehicle vs. $Apoe^{+/-}$ RGX-104). Statistical analysis was performed using Student's $t$ test (B), Mann–Whitney test (E), and one-way ANOVA test (C, D, F–J). Data are shown as mean ± SD (error bar) (*$P < 0.05$, **$P < 0.01$,***$P < 0.001$). Source data are available online for this figure.

transcriptional upregulation of APOE (Das et al, 2025). This activation impairs the survival of immunosuppressive myeloid-derived suppressor cells (MDSCs), thereby enhancing antitumor immunity (Tavazoie et al, 2018). Prior studies have highlighted the dual roles of LXRs in inflammation, which can either suppress (Bensinger et al, 2008; Zelcer et al, 2007) or promote (Fontaine et al, 2007; Waddington et al, 2015) inflammatory responses depending on the cellular context, affected cell types, and duration of LXR activation. For instance, LXR agonism has been shown to enhance macrophage-mediated phagocytosis of senescent neutrophils (Hong and Tontonoz, 2014), underscoring its context-dependent effects on myeloid cells. In the current study, we employed RGX-104 as a rescue agent to mitigate cold-induced exacerbation of OA. Concerning safety, no mortality or histopathological damage (e.g., inflammatory cell infiltration or tissue injury) was observed in major organs (heart, liver, spleen, lung, kidney) of mice treated with intra-articular RGX-104 (Appendix Fig. S5). Furthermore, RGX-104 is currently being evaluated in a Phase I clinical trial for refractory cancers (NCT02922764), which preliminarily supports its systemic safety. However, administration of RGX-104 in our study has limitations. First, the administration route in our work (intra-articular injection) differs from the oral delivery used in clinical trials, and the long-term local effects of RGX-104 on joint tissues (e.g., inflammation or tumorigenicity) require further investigation, despite the absence of adverse reactions in our histopathological analyses. Second, the observation period and sample size were limited; future studies should extend treatment duration and increase cohort sizes to evaluate long-term outcomes. Given the need to further validate RGX-104 safety, our findings suggest alternative therapeutic strategies targeting APOE-mediated lipid homeostasis. Specifically, APOE-dependent lipid efflux mitigates osteoarthritis progression, highlighting lipid metabolism regulators as safer candidates. For instance, fenofibrate, a clinically approved hypolipidemic agent, enhances HDL cholesterol and cholesterol efflux (Louati et al, 2015), mirroring protective role of APOE. Notably, fenofibrate reduces cartilage degradation in preclinical OA models (Lee et al, 2025), consistent with our mechanistic insights. Similarly, apolipoprotein A1-binding protein (AIBP), which promotes cholesterol efflux, is suppressed during OA progression, while its overexpression attenuates disease severity (Fang et al, 2013). These findings position lipid homeostasis as a druggable axis, offering actionable alternatives for combating cold-aggravated OA.

In conclusion, our study suggested that $Apoe$ is one of the main key regulatory factor in OA at low temperatures. A decreased level of APOE in low temperatures increases lipid accumulation in cartilage and results in the aggravation of OA. Therefore, targeting $Apoe$ in cartilage could be a potent treatment to control the progression of OA at low temperatures, which needs more investigation in the future.

# Methods

**Reagents and tools table**

| Reagent/resource | Reference or source | Identifier or catalog number |
| --- | --- | --- |
| **Experimental models** | | |
| C57BL/6 mice | Shanghai SLAC Laboratory Animal Co., Ltd | C57BL/6Slac |
| $Apoe^{f/f}$ mice | Jiangsu Jicuiyaokang Biotechnology Co., Ltd | T008395 |
| $Acan$-CreERT2 mice | Jiangsu Jicuiyaokang Biotechnology Co., Ltd | T055140 |
| **Antibodies** | | |
| Rabbit anti-COLII | Proteintech | 28459-1-AP |
| Rabbit anti-MMP13 | Abcam | ab39012 |
| Rabbit anti-APOE | Abcam | ab18359 |
| Rabbit anti-β-Actin | Proteintech | 20536-1-AP |
| Rabbit anti-ACAN | Proteintech | 13880-1-AP |
| Rabbit anti-Nav1.7 | Proteintech | 20257-1-AP |
| Rabbit anti-TAC1 | Proteintech | 13839-1-AP |

| Reagent/resource | Reference or source | Identifier or catalog number |
|---|---|---|
| Rabbit anti-NOX2 | Proteintech | 19013-1-AP |
| Rabbit anti-LXRβ | Abcam | ab315082 |
| **Oligonucleotides and other sequence-based reagents** | | |
| PCR Primers | This study | Table EV2 |
| **Chemicals, enzymes and other reagents** | | |
| RGX-104 | Med Chem Express | HY-111498A |
| Tamoxifen | Selleck | S1238 |
| Infinity Cholesterol Liquid Stable Reagent | Thermo Fisher Scientific | TR13421 |
| Infinity Triglycerides Liquid Stable Reagent | Thermo Fisher Scientific | TR22421 |
| Type II collagenase | Sigma-Aldrich | C2-28 |
| TRAP staining reagent | Servicebio, China | G1050-50T |
| SYBR Green | Applied Biosystems | A25780 |
| **Software** | | |
| Image J | National Institutes of Health | https://imagej.nih.gov/ |
| GraphPad Prism 7 | GraphPad | http://www.graphpad.com/ |
| Image-Pro Plus 6 | Analytical Technologies | https://analytical-online.com/ |
| Arcmap 10.7 | Environmental Systems Research Institute | https://www.esri.com |
| **Other** | | |
| HDL and LDL/VLDL Quantitation Kit | Sigma-Aldrich | MAK045 |
| DCFH-DA probe kit | Beyotime, China | S0033M |
| JC-1 assay kit | Beyotime, China | C2006 |
| TUNEL assay kit | Beyotime, China | C1090 |
| BODIPY 493/503 probe | Invitrogen | D3922 |
| High-Capacity cDNA Reverse Transcription Kit | Applied Biosystems | 4387406 |
| TSA fluorescence multiple staining kit | RecordBio, China | RC0086-23RM |

## Human subject study

Human cartilage specimens were collected from patients over 65 years old with knee osteoarthritis grade III or IV undergoing total knee arthroplasty in southern (Shanghai, Guangzhou) and northern (Dalian, Jinan, Shenyang) cities in China between December 2019 and February 2020. Exclusions were trauma, fracture, rheumatoid and infectious arthritis, neoplastic joint disease. The experiments with human samples conformed to the principles set out in the WMA Declaration of Helsinki and the Department of Health and Human Services Belmont Report. The study was approved by ethical committees of Shanghai East Hospital (Approval No. 2024163), Shandong Provincial Hospital Affiliated to Shandong First Medical University (Approval No. 2024-680), Dalian Medical University First Hospital (Approval No. 20220176) and The First Affiliated Hospital of Sun Yat-sen University (Approval No.

2023H014). Informed consent was obtained from all participating patients. Patient data including diagnosis and serum lipids were obtained. Monthly temperatures were analyzed using ArcMap 10.7 software. Knee articular temperatures were measured using a thermoprobe after 8–12 h of outdoor activity. Cartilage specimens from medial and lateral compartments were collected for histologic analysis.

## Animal studies

The animal experiments conducted in this study complied with the ARRIVE guidelines. Blinding was performed during the animal study. The study protocol followed ethical guidelines and was approved by the Ethics Committee of Shanghai East Hospital, Tongji University (Approval No. 2024072). Wild-type C57BL/6 mice were from Shanghai SLAC Laboratory Animal Co., Ltd. *Apoe*$^{f/f}$ and *Acan*-CreERT2 mice were obtained from Jiangsu Jicuiyaokang Biotechnology Co., Ltd. Conditional *Apoe* knockout *Apoe*$^{-/-}$ mice were generated by crossing *Apoe*$^{f/f}$ with *Acan*-CreERT2 mice. Tamoxifen injections (100 mg kg$^{-1}$ body weight) induced *Apoe* deletion in 8-week-old mice (Fig. 3A). PCR and sequencing confirmed genotypes. Specific primers used for genotyping are provided in Table EV2.

To study the impact of environmental temperature on knee articulation in mice, wild-type male mice were exposed to temperatures ranging from 25 to 0 °C for 24 h. Knee articular temperatures of mice were measured using a thermoprobe (Appendix Fig. S1). The thermoprobe was calibrated using the ice-point and boiling-point methods under standard atmospheric pressure. A container was filled with crushed ice, followed by distilled water to full capacity. The mixture was thoroughly stirred, and the probe was fully immersed for 30 s. The validated ice-water equilibrium temperature was 0 °C. Boiling water was poured into a container, and the probe was fully submerged for 30 s. The validated boiling temperature was 100 °C. The measurement error of the calibrated probe was confirmed to be within ±0.1 °C. Anesthetized mice were positioned supine, and a needle was inserted into the medial joint space of the knee. The thermoprobe was advanced through the cavity and positioned between the medial meniscus and medial femoral condyle. Temperature readings were recorded after a 10-s stabilization period.

Osteoarthritis was induced in 8-week-old male mice by destabilizing the knee joint's medial meniscus (DMM). Mice were then divided randomly into groups housed at 23 °C ± 1 °C and 10 °C ± 1 °C after either sham or DMM surgery. A rescue study using the LXR agonist RGX-104 (bought from Med Chem Express) explored OA exacerbation mechanisms at lower temperatures. Mice in the LT DMM + RGX-104 group received RGX-104 articular injection (4 mg/kg, twice a week, articular injection, saline as vehicle). After 8 weeks, mice were euthanized for histological analysis. Anesthesia was induced by ketamine and xylazine injection, followed by euthanasia via carbon dioxide.

Eight weeks post DMM surgery, mice knee joints were scanned using a SCANCO 50 machine. Subchondral cartilage images were reconstructed and BV/TV analyzed. Open field tests tracked mice movements in clear chambers to study behavior under different temperatures. Gait parameters of freely moving mice were analyzed using the CatWalk XT system (Noldus Information Technology). The automated platform captures parameters like dynamic paw pressure,

contact area, and heatmap patterns via a glass-floored walkway with internal illumination to visualize paw-ground interactions. Mice traversed the walkway individually, and synchronized video recordings were analyzed to quantify locomotion metrics.

## Chondrocyte isolation and culture

Murine chondrocytes were isolated from C57BL/6 mice according to the previous study (Fu et al, 2024). In short, mice were sacrificed under excessive anesthesia. Using scissors, the soft tissue and skin were cut away, and after being collected, the cartilage tissue was cleaned in PBS. Then, the cartilage tissue was cut and was digested overnight in high-glucose DMEM (Invitrogen, USA) supplemented with 0.2% type II collagenase (Sigma, USA) and 1% PS. The cell suspension was filtered with a 40-μm cell strainer and centrifuged at $400 \times g$ for 5 min. The gathered cells were again suspended in high-glucose DMEM that had 1% P/S and 10% FBS added. Every other day, the medium was changed.

Samples of human cartilage were taken from patients having total knee replacements. Specimens of cartilage were divided into severely damaged (OA) and healthy (non-OA) regions. Human cartilage tissues were cut into small pieces. The subsequent steps were consistent with the isolation of murine chondrocytes.

To establish an OA cell model at different temperatures, murine chondrocytes were treated with IL-1β (10 ng/ml) and cultured at 37 °C or 33 °C. RGX-104 (2 μM) was added for the rescue study in the 33 °C group. Human chondrocytes from OA or non-OA cartilage regions were cultured at 33 °C or 37 °C for subsequent RNA sequencing.

## Cell viability assay

The impact of varying temperatures on the IL-1β-induced OA cell model was examined using CCK8. Briefly, chondrocytes were cultured in a 96-well plate for one night before receiving a 24-h treatment of IL-1β (10 ng/ml) at either 33 °C or 37 °C. Subsequently, each well received 10 μL of CCK8, and the chondrocytes were incubated with them for 2 h at 37 °C. The absorbance at 450 nm was measured using the microplate reader.

## Chondrocyte differentiation assay

Chondrocytes were differentiated as a micromass (Zhang et al, 2023). Briefly, in a 24-well plate, droplets of primary murine chondrocytes (25,000 cells in 25 μl) were carefully seeded and then incubated for 2 h. Then, complete chondrogenic medium (DMEM/F12 supplemented with ITS, ascorbate (50 μg/mL), sodium pyruvate (1 mM), proline (40 μg/mL), dexamethasone (100 nM) and TGF-β3 (10 ng/mL), with IL-1β (10 ng/ml) was added and refreshed every other day. For seven days, the micromass was cultured at either 37 °C or 33 °C. After fixing, the micromass was stained with alcian blue. Alcian blue was used to visualize the micromass's GAG content, and Image-Pro Plus was used for analysis.

## RNA sequencing analysis

In order to examine how cold temperatures aggravate osteoarthritis, human OA and non-OA chondrocytes were cultured for five days at 37 °C and 33 °C. For every group, three biological replicates

were used. Total RNA from human non-OA and OA chondrocytes cultured at the two temperatures was extracted with TRIzol reagent. The Agilent 2100 Bioanalyzer (Agilent Technologies, Santa Clara, CA, USA) was used to evaluate the integrity of the RNA. The samples that had an RNA integrity number (RIN) of at least seven underwent further examination. The libraries were generated using the TruSeq Stranded mRNA LT Sample Prep Kit (Illumina, San Diego, CA, USA) in compliance with the manufacturer's instructions. These libraries were then sequenced using the Illumina sequencing platform (HiSeqTM 2500). The differentially expressed genes (DEGs) of the OA and non-OA chondrocytes cultured at 37 °C and 33 °C were analyzed. PCA (Principal Component Analysis) was performed to analyze the variance of expressed genes in different groups. Functional enrichment analyses of GO (Gene Ontology) and KEGG (Kyoto Gene and Genome Encyclopedia) analyses, were carried out.

Similarly, primary chondrocytes were isolated from C57BL/6 mice, including wild-type (WT, $n = 3$) and *Apoe* knockout (Apoe KO, $n = 3$) groups. Cells from each genotype were cultured under standardized conditions, followed by RNA extraction and sequencing. Genome-wide transcriptomic profiling via RNA-seq was performed to compare gene expression patterns between *Apoe* KO (Case) and WT (Control) chondrocytes, aiming to identify APOE-dependent downstream targets and regulatory pathways.

## ROS production assay

Cellular ROS determination was performed using a dichlorodihydrofluorescein diacetate (DCFH-DA) probe (Beyotime, China). In short, chondrocytes were seeded into a 24-well plate for 24 h under various conditions. The DCFH-DA probe was then added after the cells had been three times cleaned with PBS. The final concentration was 10 μM. Hoechst was used to stain the nuclei. The cells were incubated at 37 °C in the dark for 25 min, and intracellular ROS were observed using fluorescence microscopy. The wavelengths (ex/em) were 488 nm and 525 nm, respectively. The images and relative ROS intensity were analyzed using Image-Pro Plus.

## JC-1 staining

A JC-1 assay kit (Beyotime, China) was used to measure the mitochondrial membrane potential. A 24-well plate was seeded with chondrocytes (Zeng et al, 2021). The cells were treated for twenty-four hours, and then the JC-1 working solution (250 μl medium and 250 μl JC-1 stain) was added. The final concentration was 2 μM. The cells were then twice washed with PBS. After incubation at 37 °C in darkness for 20 min, the JC-1 aggregates (red) and monomers (green) were observed using fluorescence microscopy. The wavelengths (ex/em) were 490 nm/530 nm for monomers and 525 nm/590 nm for aggregates, respectively.

## TUNEL assay

Utilizing the TUNEL assay kit (Beyotime, China), cell apoptosis was measured. Chondorcytes were cultured in the 24-well plates (Wang et al, 2018). Following a 24-hour treatment period, the chondrocytes underwent a PBS wash and were then fixed using 4% paraformaldehyde. Subsequently, every well of chondrocytes were incubated with 50 μl of TUNEL working solution (containing 5 μl

TdT enzyme and 45 μl Cy3-dUTP labeling buffer) for 60 min at 37 °C in the dark. The nuclei were stained with DAPI. The apoptotic cells were then examined under a microscope. The wavelength (ex/em) were 550 nm/570 nm.

## BODIPY staining

Lipid accumulation in chondrocytes was measured using the BODIPY 493/503 (Invitrogen, USA) probe. Following a 24-h treatment period, the chondrocytes underwent a PBS wash and were then fixed for 20 min in 4% paraformaldehyde. Following a 5-min permeabilization with 0.1% Triton X-100, BODIPY 493/503 was added and left for 20 min. The final concentration was 10 μM. DAPI was used to stain the nuclei. Additionally, fixed cryosections of mouse cartilage were stained for 20 min with BODIPY 493/503, and then counterstained with DAPI. The wavelengths (ex/em) was 488 nm/525 nm.

## Safranin O-fast green staining

Pathological evaluation of cartilage in mice was performed by safranin O-fast green staining. Mice knee joints were decalcified in 10% EDTA (pH 7.4) for 21 days after being fixed in 4% paraformaldehyde for 24 h. Subsequently, the samples were sectioned into 4 μm-thick serial pieces and embedded in paraffin. A gradient series of alcohol was used to dewax and rehydrate the selected sections. After that, safranin O-fast green was used to stain the sections. The OARSI scoring system (0–6) was used to assess the degree of cartilage degeneration.

## TRAP staining

The preparation of subchondral cartilage sections was the same as above. The sections were stained with TRAP staining reagent (G1050-50T, Servicebio, Wuhan, China) as in previous studies (Song et al, 2022). The TRAP staining was used to analyse the activated osteoclasts in subchondral cartilage.

## RNA assay by RT-qPCR

Reverse transcription was performed on total RNA extracted from mice cartilage or chondrocytes using the High-Capacity cDNA Reverse Transcription Kit (Applied Biosystems, 4387406). RT-qPCR was performed in triplicate with SYBR Green (Applied Biosystems, A25780) using Applied Biosystems Real-time PCR system. mRNA levels were normalized to Gapdh and reported as relative mRNA fold change. The information of primers were listed in Table EV2.

## Western blotting

Total protein was extracted from cells with RIPA lysis buffer. The concentrations of the extracted protein samples were determined by a BCA kit. After being separated using 10% SDS-PAGE, the protein samples were moved to PVDF membranes. The blots were incubated at 4 °C overnight with antibodies against the following: collagen II (28459-1-AP, Proteintech), MMP13 (ab39012, Abcam), APOE (ab183597, Abcam), NOX2 (19013-1-AP, Proteintech), LXRβ (ab315082, Abcam) and β-actin (20536-1-AP, Proteintech). The dilution ratios of antibodies were 1:1000 for collagen II, 1:3000 for MMP13, 1:2000 for APOE, 1:2000 for NOX2, 1:1000 for LXRβ and 1:4000 for β-actin. The blots were

incubated with a goat anti-rabbit secondary antibody for 2 h following three PBS washes. The protein bands were visualized using ECL.

## Immunohistochemical staining

Immunohistochemical staining was performed to evaluate collagen II, MMP13 and APOE expression in cartilage. Cartilage sections were subjected to antigen retrieval using 0.01 M sodium citrate buffer (0.05% Tween 20, pH 6.0), which was followed by a 30-min blocking period with normal goat serum. Sections were incubated overnight at 4 °C with primary antibodies against collagen II, MMP13, and APOE. The dilution ratios for the antibodies were 1:500 for collagen II, 1:800 for MMP13, and 1:1000 for APOE. DAB was used to identify the antigen following a 30-min incubation period with a secondary antibody conjugated with peroxidase. The percentage of the positive region was visualized and evaluated using IPP.

## Immunofluorescence staining

Tissue sections of knee joints and L4 dorsal root ganglions of mice underwent heat-induced antigen retrieval in sodium citrate buffer (100 °C, 10 min), followed by PBS rinses and blocking with 1% normal goat serum. Sections were incubated with primary antibodies (Dilution ratio: 1:500 for ACAN, APOE, TAC1 and Nav1.7) at 4 °C overnight, then with species-matched secondary antibodies (1 h, room temperature). The sections were then stained with a TSA Fluorescence multiple staining kit (RC0086-23RM, Record Bio, Shanghai, China) according to the manufacturer's instructions (TYR-520 for Red and TYR-570 for Blue). Nuclei were

### The paper explained

**Problem**
Osteoarthritis (OA), a common joint disease characterized by cartilage degeneration, contributes significantly to the joint pain and joint dysfunction. It was reported that approximately two-thirds of OA patients exhibit weather sensitivity, with many identifying cold weather as a significant factor exacerbating their symptom. In terms of this, the impact of cold exposure on the progression of OA and the underlying mechanism remains inadequately addressed.

**Results**
In our study, we discovered that low-temperature exposure exacerbates cartilage degeneration in OA mice model. We identified *Apoe* as a critical protective gene that is downregulated in low-temperature conditions. The reduction in APOE levels correlates with increased lipid accumulation in cartilage, leading to chondrocyte apoptosis. Moreover, our findings indicate that *Apoe* deficiency intensifies the progression of osteoarthritis under cold exposure. Notably, the administration of the RGX-104, an LXRβ agonist, reversely restored APOE expression and reversed the deleterious effects on cold weather-sensitive OA progression.

**Impact**
Our study reveals the role of *Apoe* gene in the progression of OA in the cold conditions. Targeting on *Apoe* in cartilage may serve as a potent therapeutic strategy for managing OA progression in cold environments, particularly for OA patients residing in high-altitude or high-latitude regions, as well as during the winter season.

counterstained with DAPI. The wavelengths (ex/em) was 490 nm/520 nm for TYR-520 and 550 nm/570 nm for TYR-570.

## Statistical analysis

All tests were performed in triplicate at minimum. Data were expressed as the mean ± SD. The Shapiro–Wilk test was used to analyze normality. The equal variance test was completed by Brown–Forsythe test or Bartlett's test. Student's $t$ test was used for two group comparisons, and one-way ANOVA with Tukey's post hoc test was used for multiple group comparisons. For data which did not pass the normality test, Mann–Whitney $U$ test was used for two group comparisons, and Kruskal–Wallis test was used for multiple group comparisons. For data which did not pass the equal variance test, Welch's ANOVA test was used for multiple group comparisons. Statistical analyses were performed using GraphPad Prism 8.0 (GraphPad Software, San Diego, California, USA) and SPSS 19.0 (SPSS Inc., Chicago, IL, USA), and $P < 0.05$ was considered the criterion for a significant difference.

## Data availability

The datasets of RNA-sequencing produced in this study are available in the following databases: RNA-Seq data: Gene Expression Omnibus GSE296344. RNA-Seq data: Gene Expression Omnibus GSE296343.

The source data of this paper are collected in the following database record: biostudies:S-SCDT-10_1038-S44321-025-00268-6.

## Peer review information

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

## Acknowledgements

We sincerely appreciate the support we received from Shanghai OE Biotech Co., Ltd. and human OA samples harvest and contributed by Dr.Senbo An from Shandong Provincial Hospital Affiliated to Shandong First Medical University, Dr. Jun Hao from the First Affiliated Hospital of Dalian Medical University, Dr. Tao Yu from Guangdong Provincial People's Hospital. This work was sponsored by the National Natural Science Foundation of China (82102600, 82472504, 82102538), Shanghai Sailing Program(21YF1405800), Shanghai "Rising Stars of Medical Talents" Youth Development Program (2025-71), Health Science and Technology Project of Shanghai Pudong New Area Municipal Health Commission (PW2024B-07), Shanghai Pudong Science and Technology Development Funding (PKJ2020-Y44), The Featured Clinical Discipline Project of Shanghai Pudong New District (Pwyts2021-03 and Shandong Provincial Natural Science Foundation (ZR202102210485).

## Author contributions

**Yueqi Zhang**: Conceptualization; Resources; Data curation; Software; Formal analysis; Funding acquisition; Validation; Investigation; Visualization; Methodology; Writing—original draft; Project administration; Writing—review and editing. **Mei Fu**: Conceptualization; Resources; Data curation; Formal analysis; Validation; Investigation; Methodology; Writing—original draft; Project administration. **Chun Zhou**: Resources; Investigation; Visualization; Methodology; Writing—original draft; Project administration; Writing—review and editing. **Xiao Wang**: Software; Formal analysis; Investigation; Methodology. **Zengxin Jiang**: Data curation; Formal analysis; Validation; Investigation; Visualization; Methodology; Writing—original draft; Project administration; Writing—review and editing. **Chang Jiang**: Conceptualization; Resources; Funding acquisition; Validation; Investigation; Visualization; Methodology; Writing—original draft. **Shengyang Guo**: Resources; Software; Formal analysis; Project administration. **Zhiying Pang**: Resources; Formal analysis; Validation; Investigation; Visualization; Methodology; Project administration. **Chenzhong Wang**: Conceptualization; Supervision; Validation; Investigation; Visualization; Methodology. **Tao Yu**: Resources; Data curation; Funding acquisition; Validation; Investigation; Visualization; Methodology. **Senbo An**: Conceptualization; Resources; Data curation; Formal analysis; Supervision; Funding acquisition; Investigation; Methodology; Writing—review and editing. **Xiuhui Wang**: Conceptualization; Resources; Data curation; Formal analysis; Supervision; Funding acquisition; Validation; Writing—review and editing. **Zhe**

**Wang**: Conceptualization; Resources; Data curation; Formal analysis; Supervision; Funding acquisition; Validation; Investigation; Visualization; Methodology; Writing—review and editing.

Source data underlying figure panels in this paper may have individual authorship assigned. Where available, figure panel/source data authorship is listed in the following database record: biostudies:S-SCDT-10_1038-S44321-025-00268-6.

## Disclosure and competing interests statement

The authors declare no competing interests.

# Expanded View Figures

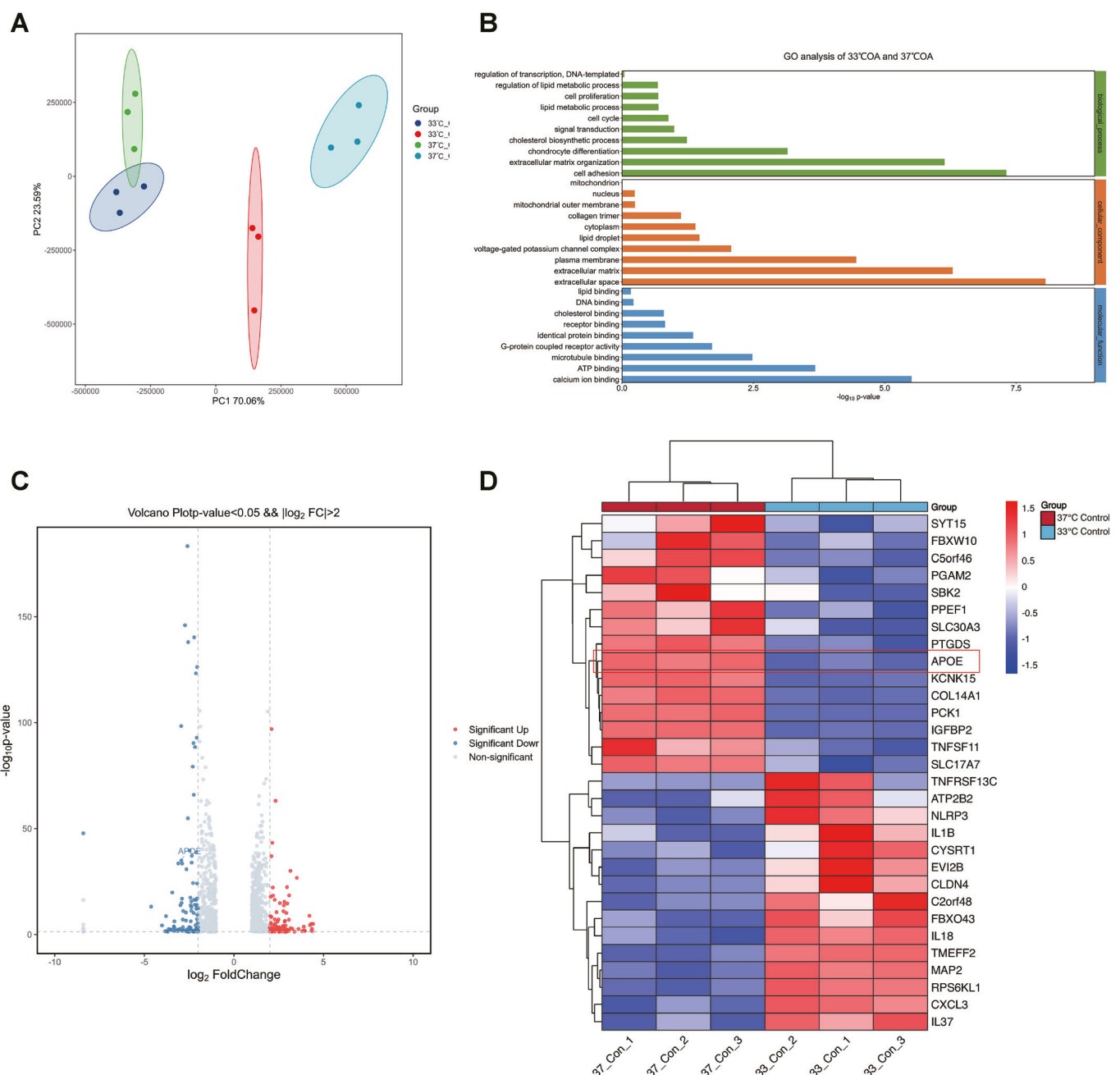

**Figure EV1. Bioinformatics analysis of RNA sequencing of chondrocytes cultured at different temperatures.**

(**A**) Principal Component Analysis (PCA) of the gene expression. (**B**) GO analysis of the DEGs. (**C**) Volcano plot of gene expressions between 33 °C control group and 37 °C control group ($n = 3$ patients per group). (**D**) Heatmap of DEGs between 33 °C control group and 37 °C control group. Statistical analysis was performed using the hypergeometric test (**B**) and the Wald test within the DESeq2 framework (**C**).

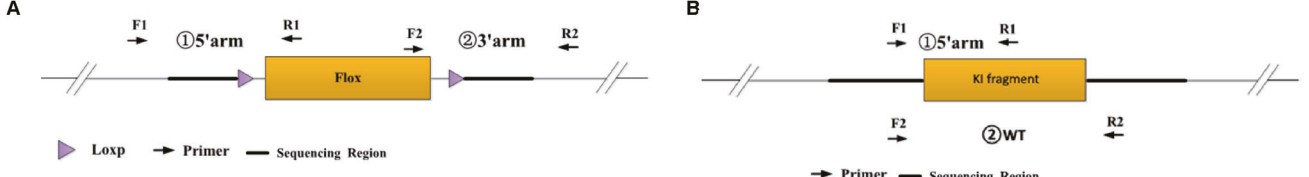

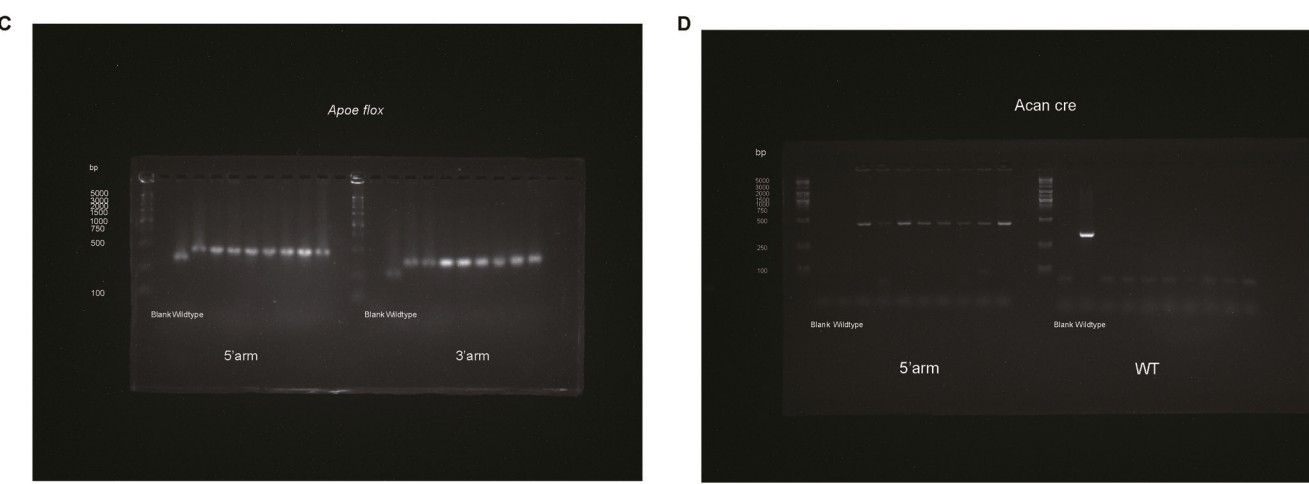

**Figure EV2. Result of the genotyping of *Apoe*<sup>−/−</sup>*Acan*<sup>Cre-ERT2</sup> mice.**

(A) Genotyping strategy for *Apoe*<sup>f/f</sup>. (B) Genotyping strategy for *Acan*<sup>Cre-ERT2</sup>. (C) Genotyping result for *Apoe*<sup>f/f</sup>. (D) Genotyping result for *Acan*<sup>Cre-ERT2</sup>.

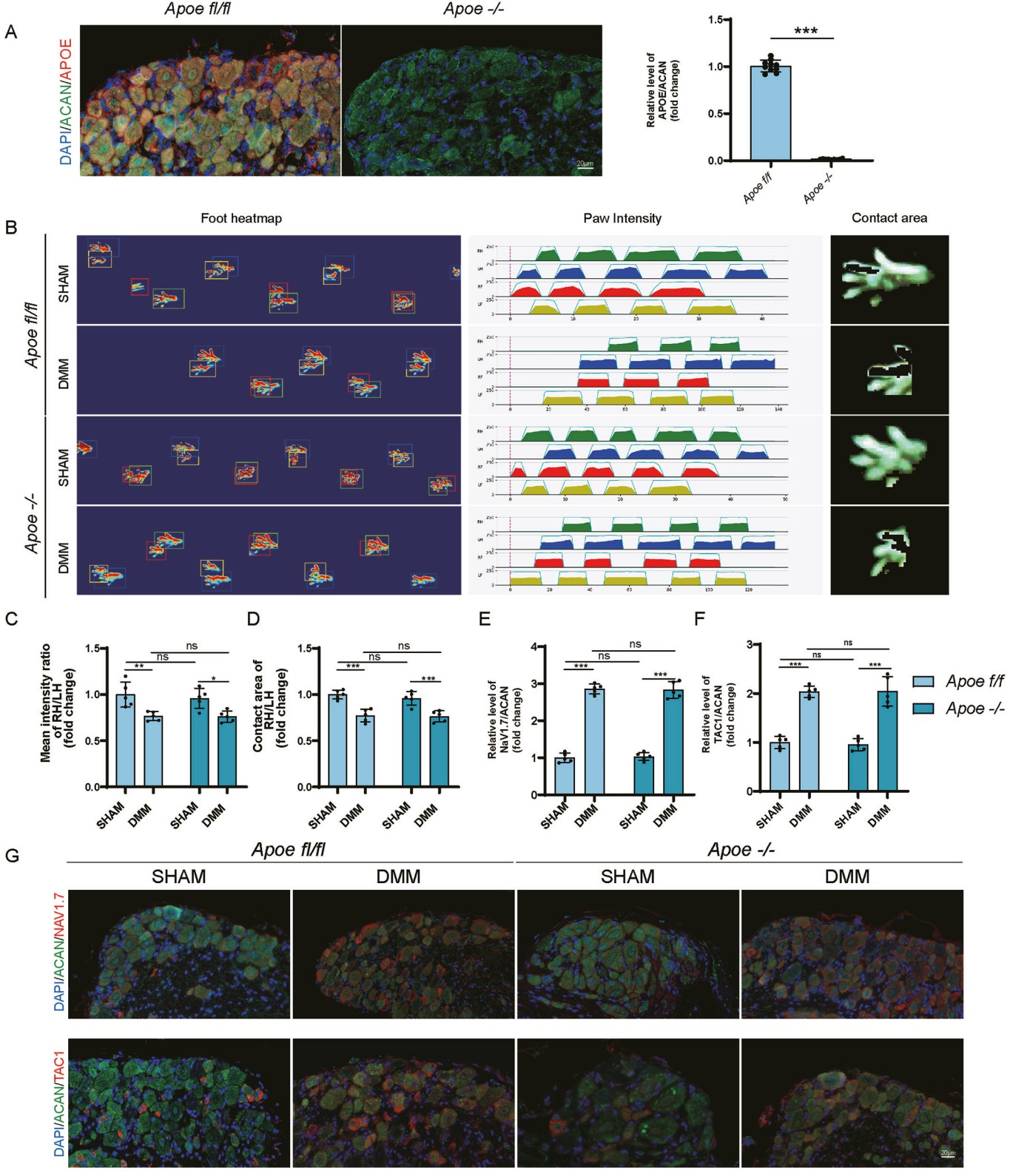

◀ **Figure EV3. Analysis of catwalk gait parameters and immunohistochemical staining for pain-related markers in dorsal root ganglia of *Apoe^{f/f}* and *Apoe^{−/−}* mice.**

(A) Immunofluorescence staining and quantitative analysis of ACAN and APOE co-expression in dorsal root ganglia ($n = 5$). $P < 0.0001$. (B) Catwalk gait analysis of *Apoe ^{f/f}* and *Apoe^{−/−}* mice and typical result of foot heatmap, paw intensity and contact area. (C) Quantitative analysis of mean intensity ratio of right hind (RH) / left hind (LH) limb ($n = 5$). $P = 0.0065$ (*Apoe^{f/f}* SHAM vs. *Apoe^{f/f}* DMM), $P = 0.0224$ (*Apoe^{−/−}* SHAM vs. *Apoe^{−/−}* DMM), $P = 0.9032$ (*Apoe^{f/f}* SHAM vs. *Apoe^{−/−}*SHAM), $P = 0.9999$ (*Apoe^{f/f}* DMM vs. *Apoe^{−/−}* DMM). (D) Quantitative analysis of ratio of contact area of RH/LH ($n = 5$). $P = 0.0002$ (*Apoe^{f/f}* SHAM vs. *Apoe^{f/f}* DMM), $P = 0.0009$ (*Apoe^{−/−}* SHAM vs. *Apoe^{−/−}* DMM), $P = 0.7479$ (*Apoe^{f/f}* SHAM vs. *Apoe^{−/−}* SHAM), $P = 0.9954$ (*Apoe^{f/f}* DMM vs. *Apoe^{−/−}* DMM). (E) Quantitative analysis of Nav1.7/ACAN level in DRG ($n = 5$). $P < 0.0001$ (*Apoe^{f/f}* SHAM vs. *Apoe^{f/f}* DMM), $P < 0.0001$ (*Apoe^{−/−}* SHAM vs. *Apoe^{−/−}* DMM), $P = 0.9863$ (*Apoe^{f/f}* SHAM vs. *Apoe^{−/−}* SHAM), $P = 0.9922$ (*Apoe^{f/f}* DMM vs. *Apoe^{−/−}* DMM). (F) Quantitative analysis of TAC1/ACAN level in DRG ($n = 5$). $P < 0.0001$ (*Apoe^{f/f}* SHAM vs. *Apoe^{f/f}* DMM), $P < 0.0001$ (*Apoe^{−/−}* SHAM vs. *Apoe^{−/−}* DMM), $P = 0.9805$ (*Apoe^{f/f}* SHAM vs. *Apoe^{−/−}* SHAM), $P = 0.9999$ (*Apoe^{f/f}* DMM vs. *Apoe^{−/−}* DMM). (G) Immunofluorescence co-staining of Nav1.7 (voltage-gated sodium channel) and TAC1 (Substance P precursor) with ACAN (Aggrecan) in dorsal root ganglia (DRG) of *Apoe^{f/f}* and *Apoe^{−/−}* mice. Statistical analysis was performed using Whelch's *t* test (A), one-way ANOVA test (C–F). Data are shown as mean ± SD (error bar) (\*$P < 0.05$, \*\*$P < 0.01$,\*\*\*$P < 0.001$). Source data are available online for this figure.

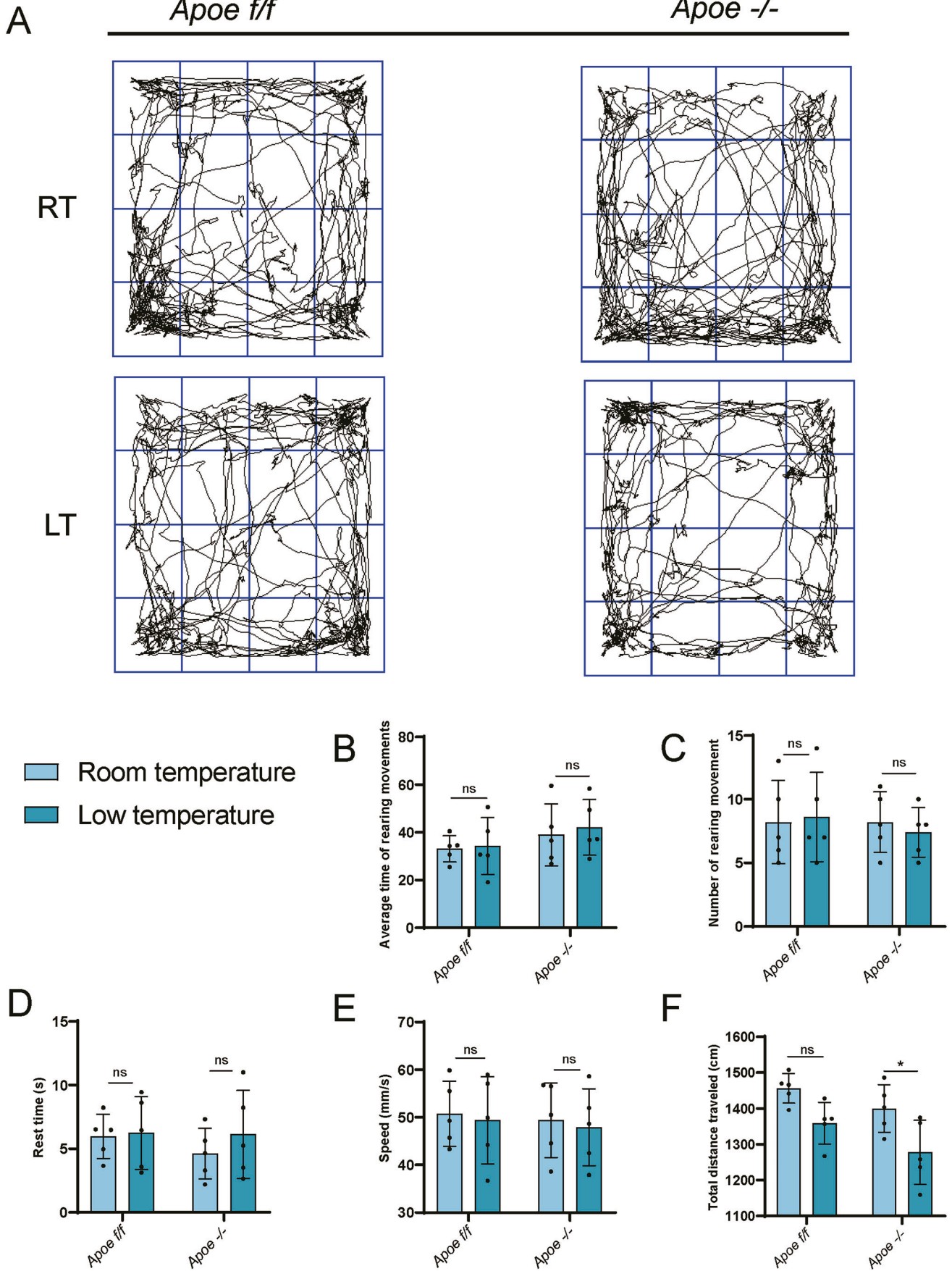

◀ **Figure EV4.  Open filed test of *Apoe*$^{f/f}$ and *Apoe*$^{-/-}$ mice at room and low temperature.**

(**A**) Images of the mouse movement pathways. (**B**) Average time of rearing movements ($n = 5$). (**C**) Number of rearing movement ($n = 5$). (**D**) Rest time ($n = 5$). (**E**) Speed ($n = 5$). (**F**) Total distance traveled ($n = 5$). $P = 0.1339$ (*Apoe*$^{f/f}$ RT vs. *Apoe*$^{f/f}$ LT), $P = 0.0449$ (*Apoe*$^{-/-}$ RT vs. *Apoe*$^{-/-}$ LT). Statistical analysis was performed using one-way ANOVA test (**B**, **D–F**), Kruskal–Wallis test (**C**). Data are shown as mean ± SD (error bar) (*$P < 0.05$). Source data are available online for this figure.

