## [Peer Review File · EMBO Molecular Medicine]

Cold exposure promotes the progression of osteoarthritis through downregulating APOE in cartilage

Yueqi Zhang, Mei Fu, Chun Zhou, Xiao Wang, Zengxin Jiang, Chang Jiang, Shenyang Guo, Zhiying Pang, Chenzhong Wang, Tao Yu, Senbo An, Xiuhui Wang, and Zhe Wang

Corresponding authors: Zhe Wang (wangzhe@renji.com) , Senbo An (ansenbo@sdfmu.edu.cn), Xiuhui Wang (zpyyqkwxh@sina.cn), Zhe Wang (wangzhe@renji.com)

Review Timeline:

Submission Date:	1st Dec 24
Editorial Decision:	4th Feb 25
Revision Received:	10th May 25
Editorial Decision:	11th Jun 25
Revision Received:	18th Jun 25
Accepted:	25th Jun 25

Editor: *Zeljko Durdevic*

Transaction Report:

4th Feb 2025

Dear Dr. Zhang,

Thank you for the submission of your manuscript to EMBO Molecular Medicine, and please accept my apologies for the unusual delay in getting back to you. We have now received feedback from two of the three reviewers who agreed to evaluate your manuscript. As the referee #3 will unfortunately not be able to return his/her report in a timely manner, and given that both reviewers provide very similar recommendations, we prefer to make a decision now in order to avoid further delay in the process.

As you will see from their reports pasted below, while the referee #1 supports publication, referee #2 recognizes interest of the study but also raise important concerns that should be addressed in a major revision. Focus of the revision should be in providing deeper mechanistic insight of cold-dependent APOE-mediated progression of osteoarthritis and better characterization of RGX-104 effects (long-term safety and detailed mechanisms of action). Should referee #3 provide a report, we will send it to you, with the understanding that we will not ask for an additional revision. If you would like to discuss further the points raised by the referees, I am available to do so via email or video. Let me know if you are interested in this option.

We would welcome the submission of a revised version within three months for further consideration. Please let us know if you require longer to complete the revision.

I look forward to receiving your revised manuscript.

Yours sincerely,

Zeljko Durdevic

We require:

- 1) A .docx formatted version of the manuscript text (including legends for main figures, EV figures and tables). Please make sure that the changes are highlighted to be clearly visible.
- 2) Individual production quality figure files as .eps, .tif, .jpg (one file per figure). For guidance, download the 'Figure Guide PDF': (<https://www.embopress.org/page/journal/17574684/authorguide#figureformat>).
- 3) A .docx formatted letter INCLUDING the reviewers' reports and your detailed point-by-point responses to their comments. As part of the EMBO Press transparent editorial process, the point-by-point response is part of the Review Process File (RPF), which will be published alongside your paper.
- 4) A complete author checklist, which you can download from our author guidelines

(<https://www.embopress.org/page/journal/17574684/authorguide#submissionofrevisions>). Please insert information in the checklist that is also reflected in the manuscript. The completed author checklist will also be part of the RPF.

6) It is mandatory to include a 'Data Availability' section after the Materials and Methods. Before submitting your revision, primary datasets produced in this study need to be deposited in an appropriate public database, and the accession numbers and database listed under 'Data Availability'. Please remember to provide a reviewer password if the datasets are not yet public (see <https://www.embopress.org/page/journal/17574684/authorguide#dataavailability>).

.

- the medical issue you are addressing,

- the results obtained and

- their clinical impact.

12) Author contributions: You will be asked to provide CRediT (Contributor Role Taxonomy) terms in the submission system. These replace a narrative author contribution section in the manuscript.

13) A Conflict of Interest statement should be provided in the main text.

14) Every published paper now includes a 'Synopsis' to further enhance discoverability. Synopses are displayed on the journal webpage and are freely accessible to all readers. They include a short stand first (maximum of 300 characters, including space) as well as 2-5 one-sentence bullet points that summarize the paper. Please write the bullet points to summarize the key NEW findings. They should be designed to be complementary to the abstract - i.e. not repeat the same text. We encourage inclusion of key acronyms and quantitative information (maximum of 30 words / bullet point). Please use the passive voice. Please attach these in a separate file or send them by email, we will incorporate them accordingly.

15) Include a Reagents and Tools Table as part of the Methods section, which can be downloaded from our author guidelines (<https://www.embopress.org/page/journal/17574684/authorguide#structuredmethods>)

***** Reviewer's comments *****

Referee #1 (Comments on Novelty/Model System for Author):

The influence of cold has not been investigated thoroughly in OA and ApoE seems to be a good candidate as mediator, however, the clinical relevance is rather low, because playing around with fat metabolism seems rather far fetched.

Referee #1 (Remarks for Author):

The manuscript is well-written and concise, the methods applied serve its purpose and the topic is interesting for the OA community. Especially the histology and IHC is good and the DMM OA model seems to have worked in this case.

Referee #2 (Comments on Novelty/Model System for Author):

The present study reveals the role and the underlying mechanism of cold climate on OA progression, and demonstrates the ApoE in cartilage as a potent therapeutic strategy for managing OA progression.

Referee #2 (Remarks for Author):

In the study, Yueqi Zhang et. al. provide novel insights into the role of low temperature in exacerbating osteoarthritis (OA) progression through APOE downregulation, shedding light on potential therapeutic targets for OA management in cold environments. The integration of patient data from different geographic regions in China strengthens the study's real-world applicability, particularly for populations in colder climates. In my opinion, this paper can be published after major revision.

What is the role of APOE in immune cells, e.g., macrophages, in the low-temperature induced OA.

ACAN gene also expressed in microglia in dorsal root, which play a key role in the development of OA and related pain. If knock down APOE in ACAN-positive microglia contribute to the OA development?

Figure 3I: please double-check the bolt of beta-actin are from the same membrane

The study uses 10{degree sign}C and 33{degree sign}C for experiments but lacks a detailed explanation of how these temperatures relate to real-world joint exposure in humans. It would be beneficial to justify the temperature choices, including whether they reflect typical cold climates experienced by OA patients.

While the study demonstrates that APOE is downregulated under cold conditions, it does not delve into the molecular pathways or transcriptional mechanisms involved in this regulation.

Adding details on potential transcription factors or signaling pathways (e.g., LXR signaling) that mediate APOE expression under cold stress would enhance the depth of the study.

The methodology for measuring joint temperature in mice is briefly mentioned but lacks technical details, such as probe calibration or potential measurement errors. Providing a more detailed description of temperature measurement protocols would improve reproducibility.

Some sentences are overly lengthy, which affects clarity. For instance, "Low temperature exacerbates OA by downregulating the

Apoe expression, leading to increased lipid accumulation..." can be streamlined to "Low temperature exacerbates OA via Apoe downregulation and lipid accumulation." Improving sentence conciseness and ensuring consistent formatting will enhance overall readability.

While the protective effects of RGX-104 are highlighted, the discussion lacks exploration of its long-term safety and detailed mechanisms of action. Including potential limitations, side effects, and alternative therapeutic strategies would provide a more balanced perspective.

This study effectively elucidates a novel mechanism linking low-temperature exposure to OA progression through APOE downregulation and lipid accumulation. It is scientifically rigorous and clinically relevant. However, improvements in temperature rationale, mechanistic exploration, and data presentation are needed. Following revisions addressing these points, the manuscript is recommended for publication.

List of Response:

Reviewer #1: I. General Comment

The influence of cold has not been investigated thoroughly in OA and ApoE seems to be a good candidate as mediator, however, the clinical relevance is rather low, because playing around with fat metabolism seems rather far fetched.

The manuscript is well-written and concise, the methods applied serve its purpose and the topic is interesting for the OA community. Especially the histology and IHC is good and the DMM OA model seems to have worked in this case.

Response: We sincerely appreciate your thoughtful review and positive feedback on our manuscript. Thank you for recognizing the significance of our work and the rigor of our methodology.

Our study was indeed motivated by clinical observations linking cold exposure to OA progression, and we aimed to explore its underlying mechanisms. Cold exposure induces lipid metabolic reprogramming through genetic and pathway modifications, as demonstrated by myocyte lipid droplet accumulation enhancing tissue oxygenation^[1]. Our findings revealed that low-temperature conditions exacerbate cartilage lipid accumulation, which correlates with reduced APOE levels. While modulating lipid metabolism may initially seem distant from OA pathology, emerging evidence increasingly positions OA as a metabolic disorder rather than a purely mechanical "wear and tear" disease. Historically, OA was attributed to repetitive joint stress^[2], but recent studies highlight its strong association with systemic metabolic conditions such as obesity, hypertension, and hyperglycemia^[3,4]. For example, Sujeong Park et al. demonstrated that PPAR α mediates lipid deposition and apoptosis in OA chondrocytes through regulation of ACOT12, a key enzyme in de novo lipogenesis^[5], underscoring the role of lipid metabolism in OA pathogenesis. Indeed, metabolic mediators, including adipokines, glucose, cholesterol, triglycerides, and their derivatives, have been shown to directly influence OA progression^[3,4,6,7]. In this context, our investigation into cold-induced lipid dysregulation and ApoE-mediated pathways aligns with the growing recognition of metabolic contributions to OA, offering novel insights into disease mechanisms that may inform future therapeutic strategies.

Reference:

[1] Sidell BD (1998) Intracellular oxygen diffusion: the roles of myoglobin and lipid at cold body temperature. *Journal of Experimental Biology* 201: 1119-1128

[2] Coaccioli S, Sarzi-Puttini P, Zis P, et al (2022) Osteoarthritis: new insight on its pathophysiology. *J Clin Med* 11:6013.

[3] Sobieh BH, El-Mesallamy HO, Kassem DH. Beyond mechanical loading: the metabolic contribution of obesity in osteoarthritis unveils novel therapeutic targets (2023) *Heliyon* 9:e15700.

[4] Wei G, Lu K, Umar M, et al. Risk of metabolic abnormalities in osteoarthritis: a new perspective to understand its pathological mechanisms (2023) *Bone Res* 11:1-16.

[5] Park S, Baek I-J, Ryu JH, Chun C-H, Jin E-J (2022) PPAR α -ACOT12 axis is

responsible for maintaining cartilage homeostasis through modulating de novo lipogenesis. *Nature Communications* 13:3

[6] Datta P, Zhang Y, Parousis A, et al. High-fat diet-induced acceleration of osteoarthritis is associated with a distinct and sustained plasma metabolite signature (2017) *Sci Rep* 7:8205.

[7] Gkretsi V, Simopoulou T, Tsezou A. Lipid metabolism and osteoarthritis: lessons from atherosclerosis (2011) *Prog Lipid Res* 50:133–140.

Reviewer #2: I. General Comment

In the study, Yueqi Zhang et. al. provide novel insights into the role of low temperature in exacerbating osteoarthritis (OA) progression through APOE downregulation, shedding light on potential therapeutic targets for OA management in cold environments. The integration of patient data from different geographic regions in China strengthens the study's real-world applicability, particularly for populations in colder climates. In my opinion, this paper can be published after major revision.

II. Specific Comment:

1. What is the role of APOE in immune cells, e.g., macrophages, in the low-temperature induced OA.

Response: Thank you for your insightful question. Emerging evidence in recent years has highlighted the pivotal role of immune regulation, particularly macrophages residing both in the synovium and subchondral bone, in the pathogenesis of osteoarthritis (OA)^[1-3]. In this study, we observed that under osteoarthritis (OA) conditions, osteoclasts in the subchondral bone exhibit aberrant activation as previously reported, leading to an imbalance over homeostasis in subchondral bone^[4]. However, with the condition of cold exposure, osteoclasts derived from the monocyte-macrophage system were not further aberrantly activated, and this imbalance did not worsen (Figure.4C and G). The absence of temperature-dependent changes in TRAP⁺ osteoclasts supports that low-temperature exposure does not exacerbate subchondral bone remodeling mediated by macrophage activity. Consequently, it is reasonable to investigate potential changes in macrophages confined within the synovium.

To investigate the relationship between synovial macrophages and OA under low-temperature conditions, as well as the role of APOE, we further performed immunofluorescence staining on synovial tissues of DMM and SHAM mice both under both room-temperature (RT) or low-temperature (LT) conditions. We co-stained for M1 (iNOS) and M2 (ARG1) macrophage markers alongside APOE. Our results revealed that DMM mice exhibited increased M1 polarization (iNOS/DAPI) compared to Sham controls, while M2 polarization (ARG1/DAPI) remained unchanged (Figure Below.A-C) Notably, APOE expression in synovial macrophages was significantly downregulated in DMM mice, evidenced by reduced APOE/iNOS and APOE/ARG1 ratios (Figure Below.A, D ,E). This aligns with prior study findings showing suppressed APOE expression in macrophages under inflammatory stimuli (e.g., LPS, TNF α , IFN γ)^[5-7]. However, in LT conditions, neither DMM nor Sham mice showed significant differences in synovial macrophage polarization (iNOS or ARG1) or APOE expression compared to RT groups.

These findings suggest that synovial macrophage APOE expression is reduced during OA-associated inflammation but remains unaltered by low-temperature exposure. This implies that APOE downregulation may contribute to OA progression in

inflammatory contexts, while LT-induced OA might involve distinct mechanisms independent of APOE modulation in macrophages. Further studies are warranted to elucidate the precise regulatory mechanisms of APOE in macrophage polarization during OA.

Figure Below. Immunofluorescence staining and analysis of macrophage and APOE expression in synovial tissues. (A). Immunofluorescence staining of iNOS and ARG1 co-stained with APOE in synovial tissues of DMM and SHAM mice under room temperature (RT) or low temperature (LT) conditions. (B). Quantitative analysis of iNOS/DAPI. (C). Quantitative analysis of ARG1/DAPI. (D). Quantitative analysis of APOE/iNOS. (E). Quantitative analysis of APOE/ARG1 (* $p < 0.05$, ** $p < 0.01$, *** $p < 0.001$).

Reference:

[1] Li H, Yuan Y, Zhang L, Xu C, Xu H, Chen Z (2024) Reprogramming Macrophage Polarization, Depleting ROS by Astaxanthin and Thioketal-Containing Polymers Delivering Rapamycin for Osteoarthritis Treatment. *Advanced Science* 11: 2305363.

[2] Zhang H, Lin C, Zeng C, Wang Z, Wang H, Lu J, Liu X, Shao Y, Zhao C, Pan J et al (2018) Synovial macrophage M1 polarisation exacerbates experimental osteoarthritis partially through R-spondin-2. *Annals of the Rheumatic Diseases* 77: 1524-1534.

[3] Harasymowicz NS, Harissa Z, Rashidi N, Lenz K, Tang R, Guilak F Injury and obesity differentially and synergistically induce dysregulation of synovial immune cells in osteoarthritis. *Annals of the Rheumatic Diseases*.

[4] Zhu S, Zhu J, Zhen G, Hu Y, An S, Li Y, Zheng Q, Chen Z, Yang Y, Wan M et al (2019) Subchondral bone osteoclasts induce sensory innervation and osteoarthritis pain. *The Journal of Clinical Investigation* 129(3):1076-1093.

[5] Werb Z, Chin JR, Takemura R, Oropeza RL, Bainton DF, Stenberg P, Taylor JM,

Reardon C (1986) The Cell and Molecular Biology of Apolipoprotein E Synthesis by Macro Phages. In: *Ciba Foundation Symposium 118 - Biochemistry of Macrophages*, pp. 155-171.

[6] Duan H, Li Z, Mazzone T (1995) Tumor necrosis factor-alpha modulates monocyte/macrophage apoprotein E gene expression. *The Journal of Clinical Investigation* 96: 915-922.

[7] Brand K, Mackman N, Curtiss LK (1993) Interferon-gamma inhibits macrophage apolipoprotein E production by posttranslational mechanisms. *The Journal of Clinical Investigation* 91: 2031-2039.

2. ACAN gene also expressed in microglia in dorsal root, which play a key role in the development of OA and related pain. If knock down APOE in ACAN⁺ microglia contribute to the OA development?

Response:

Thank you for raising this important question. ACAN is indeed expressed in microglia within the dorsal root ganglia (DRG). To investigate whether *ApoE* knockdown in ACAN⁺ microglia contributes to osteoarthritis (OA) development and pain, we analyzed DRG tissues from *ApoE* conditional knockout (*ApoE* ^{-/-}) and (*ApoE* *fl/fl*) littermates. Immunostaining confirmed successful *ApoE* deletion in ACAN⁺ DRG microglia of *ApoE* ^{-/-} mice (Figure Below.A) .

To assess pain-related behavioral changes, CatWalk gait analysis revealed that DMM mice exhibited reduced paw intensity and contact area in the affected limb compared to sham controls, consistent with OA-associated pain. However, *ApoE*^{-/-} mice showed no significant differences in these parameters versus *ApoE*^{fl/fl} littermates, indicating that *ApoE* deletion in ACAN⁺ microglia does not alter pain-related behaviors (Figure Below.B-D) .Further, our data demonstrated that DMM mice exhibited upregulated expression of Nav1.7 (a sodium channel linked to OA progression and pain) and TAC1 (encoding substance P, a key pain-associated neurotransmitter) in the DRG level(Figure Below.E-G). These findings align with established mechanisms where Nav1.7 sensitizes nociceptive neurons and substance P amplifies pain signaling in OA^[1,2]. However, conditional knockout of *ApoE* in ACAN⁺ DRG microglia did not alter Nav1.7 or TAC1 expression levels, suggesting that conditional knockout of *ApoE* in these cells does not modulate these specific pain-related pathways during OA development.

While prior studies have utilized ACAN as a chondrocyte-specific driver for conditional knockouts^[3-5], our findings suggest that APOE in DRG microglia may not play a critical role in OA-associated pain or pathology under the tested conditions. We appreciate your insightful query, which has strengthened the rigorosity of our conclusions.

Changes in text: The revision was shown in the results section, line 229-247.

Figure Below. Analysis of catwalk gait parameters and immunohistochemical staining for pain-related markers in dorsal root ganglia of *Apoe f/f* and *Apoe -/-* mice. (A) Immunofluorescence staining and quantitative analysis of ACAN and APOE co-expression in dorsal root ganglia. (B) Catwalk gait analysis of *Apoe f/f* and *Apoe -/-* mice and typical result of foot heatmap, paw intensity and contact area. (C) Quantitative analysis of mean intensity ratio of right hind (RH) / left hind (LH) limb. (D) Quantitative analysis of ratio of contact area of RH/LH. (E) Quantitative analysis of Nav1.7/ACAN level in DRG. (F) Quantitative analysis of TAC1/ACAN level in DRG. (G) Immunofluorescence co-staining of Nav1.7 (voltage-gated sodium channel) and TAC1 (Substance P precursor) with ACAN (Aggrecan) in dorsal root ganglia (DRG) of *Apoe f/f* and *Apoe -/-* mice (*p<0.05,**p<0.01,***p<0.001).

Reference:

- [1] Fu W, Vasylyev D, Bi Y, Zhang M, Sun G, Khleborodova A, Huang G, Zhao L, Zhou R, Li Y et al (2024) Nav1.7 as a chondrocyte regulator and therapeutic target for osteoarthritis. *Nature* 625: 557-565.
- [2] Malek N, Mlost J, Kostrzewa M, Rajca J, Starowicz K (2024) Description of Novel Molecular Factors in Lumbar DRGs and Spinal Cord Factors Underlying Development of Neuropathic Pain Component in the Animal Model of Osteoarthritis. *Molecular Neurobiology* 61: 1580-1592.
- [3] Gong Z, Zhu J, Chen J, Feng F, Zhang H, Zhang Z, Song C, Liang K, Yang S, Fan S et al (2023) CircRREB1 mediates lipid metabolism related senescent phenotypes in chondrocytes through FASN post-translational modifications. *Nature Communications* 14: 5242.
- [4] Ng JQ, Jafarov TH, Little CB, Wang T, Ali AM, Ma Y, Radford GA, Vrbanac L, Ichinose M, Whittle S et al (2023) Loss of Grem1-lineage chondrogenic progenitor cells causes osteoarthritis. *Nature Communications* 14: 6909
- [5] Yang R, Cao D, Suo J, Zhang L, Mo C, Wang M, Niu N, Yue R, Zou W (2023) Premature aging of skeletal stem/progenitor cells rather than osteoblasts causes bone loss with decreased mechanosensation. *Bone Research* 11: 35

3. Figure 3I: please double-check the blot of beta-actin are from the same membrane.

Result of western blot of COLII and MMP13 of the chondrocytes.

Response:

Thank you for your comment. We have double checked the Western blot results in Figure 3I, reproduced the experiments, and provided the original blot images for the verification.

Changes in text: Fig. 3I and J.

4. The study uses 10°C and 33°C for experiments but lacks a detailed explanation of how these temperatures relate to real-world joint exposure in humans. It would be beneficial to justify the temperature choices, including whether they reflect typical cold climates experienced by OA patients.

Response: Thank you for raising this important methodological consideration. We appreciate the opportunity to clarify the rationale behind our temperature selections (10°C and 33°C) in relation to human joint exposure conditions. Our experimental temperature settings were carefully determined through the following key lines of evidence.

Firstly, we conducted intra-articular temperature measurements in knee osteoarthritis (OA) patients from northern and southern China during winter (December 2019-February 2020). Recorded average ambient temperatures in northern vs. southern Chinese cities were $-0.1^{\circ}\text{C} \pm 3.92^{\circ}\text{C}$ vs. $11.10^{\circ}\text{C} \pm 4.38^{\circ}\text{C}$. Patients were instructed to engage in outdoor activities for 8-12 hours prior to undergoing knee joint puncture, during which intra-articular temperature measurements were obtained using a thermoprobe. Interestingly, the latitude-temperature trend plot revealed that ambient temperature gradually decreases with increasing latitude, while articular temperature shows a declining trend that stabilizes around 33°C. Using 34°N as the demarcation, articular temperatures in patients measured $33.67^{\circ}\text{C} \pm 0.99^{\circ}\text{C}$ vs. $36.33^{\circ}\text{C} \pm 0.96^{\circ}\text{C}$ (northern vs. southern, $p < 0.001$) respectively (Figure Below.A and B). The 33°C measurement from northern patients was therefore selected as representative of cold exposure conditions in human joints.

Secondly, our temperature differential selection (33°C vs. 37°C) aligns with previous studies demonstrating that brief cold air exposure (5 minutes) induces approximately 3.9°C reduction in human knee joint temperature^[1]. This consistency with established physiological responses strengthens the clinical relevance of our experimental design. Thirdly, our murine experiments revealed a critical environmental temperature threshold: as ambient temperature decreased from 25°C to 10°C, intra-articular temperature progressively declined to 33°C, stabilizing at this level despite further environmental cooling to 0°C (Figure.1B). It was observed that mice could not tolerate ambient temperatures below 0°C for extended periods. The plateau phenomenon explains our selection of 10°C as the environmental cold exposure parameter to achieve clinically relevant joint temperatures.

By integrating human clinical data, existing literature evidence, and translational animal model findings, we established 33°C (intra-articular cold exposure) and 10°C (environmental cold exposure) as physiologically and clinically meaningful experimental conditions that reflect real-world cold exposure scenarios for OA patients.

Changes in the text: The revision was shown in the results section, line 193-202.

Figure Below. Latitude-temperature trend plot. (A) Latitude-temperature trend plot showing the relationship between knee temperature in osteoarthritis (OA) patients, ambient environmental temperature, and geographic latitude across northern and southern regions of China. (B) Statistical analysis of articular temperature (**p<0.001).

Reference:

Kim YH, Baek SS, Choi KS, Lee SG, Park SB (2002) The effect of cold air application on intra-articular and skin temperatures in the knee. *Yonsei Medical Journal*. 43(5):621-626.

5. While the study demonstrates that APOE is downregulated under cold conditions, it does not delve into the molecular pathways or transcriptional mechanisms involved in this regulation. Adding details on potential transcription factors or signaling pathways (e.g., LXR signaling) that mediate APOE expression under cold stress would enhance the depth of the study.

Response:

We sincerely appreciate the reviewer for their valuable suggestion to explore the molecular pathways underlying cold-mediated APOE regulation. Below, we provide detailed explanations of our findings related to transcriptional mechanisms, particularly the upstreaming role of Liver X receptors (LXRs) and its downstream pathways.

Liver X receptors (LXR β and LXR α), members of the nuclear hormone receptor family, are key transcriptional activators of APOE and other lipid metabolism-related genes, primarily regulating lipid efflux^[1,2]. Previous studies have shown that pharmacological activation of ubiquitously expressed LXR β (e.g., via the agonist RGX-104 used in this study) significantly upregulates APOE expression and suppresses melanoma tumor progression^[3-5]. To investigate whether cold stress regulates APOE through LXR β , we conducted in vitro and in vivo experiments.

In IL-1 β -induced chondrocytes cultured at 33°C (cold stress) versus 37°C, no significant differences in LXR β expression were observed, whereas APOE expression was markedly reduced (Figure below. A-C). RNA sequencing of human OA chondrocytes under 33°C vs. 37°C conditions showed no significant cold-induced changes in LXR β expression (p=0.108). Interestingly, treatment of 33°C-cultured *ApoE f/f* and *ApoE +/-* chondrocytes with RGX-104 increased LXR β expression, restored APOE expression, rescued COLII, suppressed MMP13 expression and decreased NOX2 expression (a key regulator of ROS production) (Figure below. A-D), indicating that LXR β activation protects chondrocyte phenotype under cold stress via activating APOE.

In the LT DMM mouse model, cartilage LXR β expression showed no significant difference compared to the RT DMM (room temperature) group. However, intra-articular injection of RGX-104 elevated LXR β expression in both groups (Figure below. D and F). These results suggest that cold stress downregulates APOE independently of LXR β expression, yet LXR β activation via RGX-104 remains a potent compensatory strategy to counteract cold-induced APOE suppression and chondrocyte dysfunction.

Our study revealed that cold stress reduces chondrocyte APOE expression, exacerbates lipid accumulation, and increases ROS production. To dissect the downstream pathways, we performed RNA sequencing on wild-type (WT) and APOE-knockout (KO) chondrocytes, identifying 384 differentially expressed genes

($p < 0.05$ and $|\log_2 \text{Foldchange}| > 1$) (Figure below.H). Gene Set Enrichment Analysis (GSEA) highlighted significant upregulation of ROS-related pathways in APOE-KO cells (Figure below.I). The ROS-related genes, such as *Cybb*, *Ncf1*, *Ncf2*, *Ncf4*, *Nos2*, *Nos3*, were markedly elevated in *ApoE* KO chondrocytes (Figure below.J). Among these genes, *Cybb* encodes NOX2, a core catalytic subunit of the NADPH oxidase (NOX) family. The NOX family represents the only enzyme class primarily dedicated to reactive oxygen species (ROS) generation, playing critical roles in redox signaling and oxidative stress regulation^[6]. Western blot analysis revealed that NOX2 expression in chondrocytes was markedly elevated in the 33°C group compared to the 37°C group, with even higher levels observed in *ApoE*^{+/-} cells (Figure below. A-C). This suggests that *ApoE* deficiency may promote NOX2 overexpression, potentially linking it to enhanced ROS production. Notably, RGX-104 treatment downregulated NOX2 expression in both cell types. Consistent with cellular findings, animal experiments demonstrated significantly increased NOX2 expression in the cartilage of LT-treated DMM mice compared to the RT group, while RGX-104 intervention effectively reduced NOX2 levels in LT mice (Figure below. E and G). These results collectively indicate that *ApoE* deficiency exacerbates NOX2-mediated ROS production under hypothermic conditions, and pharmacological intervention with RGX-104 demonstrates therapeutic potential by suppressing NOX2 expression. These findings suggest that APOE downregulation under cold stress promotes lipid accumulation, which in turn activates ROS-generating pathways through mechanisms independent of direct LXR β transcriptional regulation.

Consistent with our findings, previous studies have established NOX2 as a key mediator of oxidative damage in OA pathogenesis. Under physiological conditions, NOX2 exhibits minimal expression in healthy chondrocytes, yet its levels are dramatically elevated in OA cartilage and IL-1 β -stimulated chondrocytes^[7]. Moreover, genetic ablation of NOX2 in collagenase-induced OA models significantly attenuated disease progression, including synovial hyperplasia, osteophyte formation, and structural cartilage degeneration^[7]. Importantly, therapeutic targeting of NOX2 activity-via siRNA-mediated silencing of its essential cytosolic subunit p47phox using poly(lactic-co-glycolic) acid nanoparticles-not only suppressed ROS overproduction but also ameliorated pain-related behaviors and cartilage degradation in mono-iodoacetate-induced OA^[8]. These collective evidences underscore a ROS-dependent axis wherein NOX2 activation exacerbates OA pathology, aligning with our observation that cold stress amplifies NOX2-driven ROS generation through APOE downregulation.

Overall, while LXR β activation (via RGX-104) effectively rescues cold-induced APOE suppression and chondrocyte damage, cold stress itself regulates APOE independently of LXR β expression. The ROS overproduction observed under cold conditions likely arises from activation of downstream genes related to ROS production. We fully agree that further investigation into these compensatory pathways will deepen our understanding of cold exposure-mediated metabolic

dysregulation. We are grateful for the reviewer's insightful comments, which have strengthened the mechanistic focus of our study.

Changes in the text: The revision was shown in the results and discussion section, line307-323, line 327-330, line 332-345, line 353-359, line 367-401, line 454-490.

Figure Below. Upstream and Downstream Mechanisms of APOE Downregulation-Mediated Cold Exposure Aggravation in Osteoarthritis. (A) The western blot result of COLII, MMP13, APOE, LXR β , NOX2 in Apoef/f and ApoE^{+/-} chondrocytes. (B) The analysis of protein expression in ApoE f/f chondrocytes. (C) The analysis of protein expression in ApoE^{+/-} chondrocytes. (D) LXR β staining in cartilage. (E) Immunohistochemistry staining of NOX2 in cartilage. (F) Statistical analysis of LXR β expression. (G) Statistical analysis of NOX2 expression. (H) Volcano plot illustrating differentially expressed genes (DEGs) between wildtype (WT) and ApoE knockout (ApoE KO) chondrocytes. (I) Gene Set Enrichment Analysis (GSEA) results comparing wildtype (WT) and ApoE knockout (ApoE KO) chondrocytes, focusing on reactive oxygen species (ROS)-related pathways. (J) Heatmap depicting differentially expressed genes associated with ROS generation. (K) Bar plot showing expression levels of key differentially upregulated genes associated with ROS production in chondrocytes. (L) Schematic diagram of the overall mechanistic hypothesis of this study (*p<0.05,**p<0.01,***p<0.001).

Reference:

- [1] Evans Ronald M, Mangelsdorf David J (2014) Nuclear Receptors, RXR, and the Big Bang. *Cell* 157: 255-266.
- [2] Hong C, Tontonoz P (2014) Liver X receptors in lipid metabolism: opportunities for drug discovery. *Nature Reviews Drug Discovery* 13: 433-444.
- [3] Tavazoie MF, Pollack I, Tanqueco R, Ostendorf BN, Reis BS, Gonsalves FC, Kurth I, Andreu-Agullo C, Derbyshire ML, Posada J et al (2018) LXR/ApoE Activation Restricts Innate Immune Suppression in Cancer. *Cell* 172: 825-840.e818.
- [4] Pencheva N, Buss Colin G, Posada J, Merghoub T, Tavazoie Sohail F (2014) Broad-Spectrum Therapeutic Suppression of Metastatic Melanoma through Nuclear Hormone Receptor Activation. *Cell* 156: 986-1001.
- [5] Pencheva N, Tran H, Buss C, Huh D, Drobnjak M, Busam K, Tavazoie Sohail F (2012) Convergent Multi-miRNA Targeting of ApoE Drives LRP1/LRP8-Dependent Melanoma Metastasis and Angiogenesis. *Cell* 151: 1068-1082.
- [6] Vermot A, Petit-Härtlein I, Smith SME, Fieschi F (2021) NADPH Oxidases (NOX): An Overview from Discovery, Molecular Mechanisms to Physiology and Pathology. *Antioxidants* 10: 890.
- [7] Kruisbergen NNL, Di Ceglie I, van Gemert Y, Walgreen B, Helsen MMA, Slöetjes AW, Koenders MI, van de Loo FAJ, Roth J, Vogl T et al (2021) Nox2 Deficiency Reduces Cartilage Damage and Ectopic Bone Formation in an Experimental Model for Osteoarthritis. *Antioxidants* 10 : 1660.
- [8] Shin HJ, Park H, Shin N, Kwon HH, Yin Y, Hwang J-A, Kim SI, Kim SR, Kim S, Joo Y et al (2020) p47phox siRNA-Loaded PLGA Nanoparticles Suppress ROS/Oxidative Stress-Induced Chondrocyte Damage in Osteoarthritis. *Polymers* 12: 443.

6. The methodology for measuring joint temperature in mice is briefly mentioned but lacks technical details, such as probe calibration or potential measurement errors. Providing a more detailed description of temperature measurement protocols would improve reproducibility.

Response:

Thank you for raising this important point. We have revised the manuscript to include a more detailed description of the joint temperature measurement methodology to ensure reproducibility. Below is the expanded protocol:

Knee articular temperatures of mice were measured using a thermoprobe. The thermoprobe was calibrated using the ice-point and boiling-point methods under standard atmospheric pressure. A container was filled with crushed ice, followed by distilled water to full capacity. The mixture was thoroughly stirred, and the probe was fully immersed for 30 seconds. The validated ice-water equilibrium temperature was 0°C. Boiling water was poured into a container, and the probe was fully submerged for 30 seconds. The validated boiling temperature was 100°C. The measurement error of the calibrated probe was confirmed to be within $\pm 0.1^\circ\text{C}$. Anesthetized mice were positioned supine, and a needle was inserted into the medial joint space of the knee. The thermoprobe was advanced through the cavity and positioned between the medial meniscus and medial femoral condyle. Temperature readings were recorded after a 10-second stabilization period.

Changes in the text: The revision was shown in the methods section, line 574-585.

7. Some sentences are overly lengthy, which affects clarity. For instance, "Low temperature exacerbates OA by downregulating the Apoe expression, leading to increased lipid accumulation..." can be streamlined to "Low temperature exacerbates OA via Apoe downregulation and lipid accumulation." Improving sentence conciseness and ensuring consistent formatting will enhance overall readability.

Response:

Thank you for your valuable feedback on improving the clarity and conciseness of our manuscript. We have carefully revised the highlighted sentence and streamlined its structure to enhance readability.

The original sentence: "Low temperature exacerbates OA by downregulating the Apoe expression, leading to increased lipid accumulation..." has been revised to: "Low temperature exacerbates OA via Apoe downregulation and lipid accumulation."

We have thoroughly reviewed the entire text to address similar instances of overly lengthy phrasing and ensured consistent formatting throughout the manuscript. These revisions aim to improve scientific precision while maintaining methodological clarity.

Your constructive suggestions have significantly strengthened the quality of our work, and we sincerely appreciate your expertise in guiding these improvements.

8. While the protective effects of RGX-104 are highlighted, the discussion lacks exploration of its long-term safety and detailed mechanisms of action. Including potential limitations, side effects, and alternative therapeutic strategies would provide a more balanced perspective.

Response:

We sincerely appreciate the reviewer's thoughtful critique and the opportunity to address these important concerns. Regarding the mechanisms of RGX-104, it functions as a potent LXR β agonist that activates the LXR/ApoE axis, driving transcriptional upregulation of apolipoprotein E (ApoE)^[1]. This activation impairs the survival of immunosuppressive myeloid-derived suppressor cells (MDSCs), thereby enhancing antitumor immunity^[2]. Prior studies have highlighted the dual roles of LXRs in inflammation, which can either suppress^[3,4] or promote^[5] inflammatory responses depending on the cellular context, affected cell types, and duration of LXR activation^[5,6]. For instance, LXR agonism has been shown to enhance macrophage-mediated phagocytosis of senescent neutrophils^[7], underscoring its context-dependent effects on myeloid cells. In our study, we observed that cold exposure exacerbates osteoarthritis by downregulating ApoE expression independently of the LXR pathway, while intra-articular RGX-104 administration alleviates osteoarthritis by activating LXR β and restoring ApoE levels, thereby improving lipid efflux and reducing oxidative stress in joint cartilage.

Concerning safety, no mortality or histopathological damage (e.g., inflammatory cell infiltration or tissue injury) was observed in major organs (heart, liver, spleen, lung, kidney) of mice treated with intra-articular RGX-104 (Figure below. A). Furthermore, RGX-104 is currently being evaluated in a Phase I clinical trial for refractory cancers (NCT02922764), which preliminarily supports its systemic safety. However, our study has limitations. First, the administration route in our work (intra-articular injection) differs from the oral delivery used in clinical trials, and the long-term local effects of RGX-104 on joint tissues (e.g., inflammation or tumorigenicity) require further investigation, despite the absence of adverse reactions in our histopathological analyses. Second, the observation period and sample size were limited; future studies should extend treatment duration and increase cohort sizes to evaluate long-term outcomes.

The reviewer's suggestion to explore alternative therapeutic strategies is well-noted. Our findings indicate that ApoE-mediated lipid efflux plays a central role in mitigating osteoarthritis, suggesting that lipid metabolism regulators could serve as safer alternatives. For example, fenofibrate, a clinically approved hypolipidemic drug, elevates HDL cholesterol levels and enhances cholesterol efflux^[8], mimicking the protective effects of ApoE. Lee et al. demonstrated that fenofibrate significantly reduces cartilage degradation in osteoarthritis models^[9], aligning with our

observations. Additionally, apolipoprotein A1-binding protein (AIBP), which facilitates cholesterol efflux^[10], is downregulated during osteoarthritis progression, and its overexpression has been shown to suppress disease development^[9]. These findings reinforce lipid homeostasis as a promising therapeutic target and highlight potential alternatives to RGX-104.

In conclusion, we thank the reviewer for raising these critical points. Moving forward, we will prioritize long-term safety assessments of RGX-104 (including systemic and localized effects), further elucidate the context-dependent mechanisms of LXR activation, and evaluate alternative strategies such as fenofibrate. These efforts will provide a more comprehensive understanding of RGX-104's therapeutic potential and limitations. We deeply value the reviewer's insights, which will significantly strengthen our work.

Changes in the text: The revision was shown in the discussion section, line 503-537.

Figure Below. Evaluation of the safety of articular injection of RGX-104. (A) Representative hematoxylin and eosin (H&E)-stained sections of heart, liver, spleen, lung, and kidney from mice treated with vehicle control (n=10) or RGX-104 (n=10) articular injection.

Reference

- [1] Das S, Parigi SM, Luo X, Fransson J, Kern BC, Okhovat A, Diaz OE, Sorini C, Czarnewski P, Webb AT et al (2025) Liver X receptor unlinks intestinal regeneration and tumorigenesis. *Nature* 637: 1198-1206.
- [2] Tavazoie MF, Pollack I, Tanqueco R, Ostendorf BN, Reis BS, Gonsalves FC, Kurth I, Andreu-Agullo C, Derbyshire ML, Posada J et al (2018) LXR/ApoE Activation Restricts Innate Immune Suppression in Cancer. *Cell* 172: 825-840.e818.
- [3] Bensinger SJ, Bradley MN, Joseph SB, Zelcer N, Janssen EM, Hausner MA, Shih R, Parks JS, Edwards PA, Jamieson BD et al (2008) LXR Signaling Couples Sterol Metabolism to Proliferation in the Acquired Immune Response. *Cell* 134: 97-111.
- [4] Zelcer N, Khanlou N, Clare R, Jiang Q, Reed-Geaghan EG, Landreth GE, Vinters HV, Tontonoz P (2007) Attenuation of neuroinflammation and Alzheimer's disease pathology by liver x receptors. *Proceedings of the National Academy of Sciences* 104: 10601-10606.

- [5] Fontaine C, Rigamonti E, Nohara A, Gervois P, Teissier E, Fruchart J-C, Staels B, Chinetti-Gbaguidi G (2007) Liver X Receptor Activation Potentiates the Lipopolysaccharide Response in Human Macrophages. *Circulation Research* 101: 40-49.
- [6] Waddington Kirsty E, Jury Elizabeth C, Pineda-Torra I (2015) Liver X receptors in immune cell function in humans. *Biochemical Society Transactions* 43: 752-757.
- [7] Hong C, Kidani Y, A-Gonzalez N, Phung T, Ito A, Rong X, Ericson K, Mikkola H, Beaven SW, Miller LS et al (2012) Coordinate regulation of neutrophil homeostasis by liver X receptors in mice. *The Journal of Clinical Investigation* 122: 337-347.
- [8] Nogueira-Recalde U, Lorenzo-Gómez I, Blanco FJ, Loza MI, Grassi D, Shirinsky V, Shirinsky I, Lotz M, Robbins PD, Domínguez E et al (2019) Fibrates as drugs with senolytic and autophagic activity for osteoarthritis therapy. *eBioMedicine* 45: 588-605.
- [9] Lee G, Yang J, Kim S-J, Tran T-T, Lee SY, Park KH, Kwon S-H, Chung K-H, Koh J-T, Huh YH et al (2025) Enhancement of Intracellular Cholesterol Efflux in Chondrocytes Leading to Alleviation of Osteoarthritis Progression. *Arthritis & Rheumatology* 77: 151-162.
- [10] Fang L, Choi S-H, Baek JS, Liu C, Almazan F, Ulrich F, Wiesner P, Taleb A, Deer E, Pattison J et al (2013) Control of angiogenesis by AIBP-mediated cholesterol efflux. *Nature* 498: 118-122.

11th Jun 2025

Dear Dr. Wang,

Thank you for the submission of your revised manuscript to EMBO Molecular Medicine. I am pleased to inform you that we will be able to accept your manuscript pending the following final amendments:

1) We note that you currently have, a total of 3 first authors. Is that correct? Do you confirm equal contribution of these authors, able to take full responsibility for the paper and its content? While there is no limit per se to the number of first authors, 3 authors is rather rare, and may not reflect as intended to the community.

2) Figures:

- Please update figures 2, 3, 5, 6 and EV3 and their corresponding source data files with the revised versions.

3) In the main manuscript file, please do the following:

- Please address all comments suggested by our data editors listed below:

o Data availability statement:

1. Please note that the specific URLs for GSE296344, GSE296343 datasets are not provided in the data availability statement.

o Figure legends:

1. Please note that the exact p values are not provided in the legends of figures 1H, M, N, O, P, Q, R, S; 2G, H, L, M, N, O, P; 3B, D, E, F, G, H, J; 4E, F, G, H, I, J, K, L, P; 5C, D, J, K, L, M, N; 6B, C, D, E, F, G, H, I, J; 6B-J; EV3 A, C, D, E, F; EV4 F.

2. Please indicate the statistical test used for data analysis in the legends of figures 2A, EV1 B, C. Add callouts for Figure 4G.

3. Please note that information related to n is missing in the legends of figures 2A, EV1 C, S2, S3 A-D; S4A-E.

- There are callouts for a "Table EV" and for a Table S2 - please correct to the Table EV1 or Table EV2.

- Remove all figures and tables.

- Author contributions: Please remove it from the manuscript and specify author contributions in our submission system. CRediT has replaced the traditional author contributions section because it offers a systematic machine-readable author contributions format that allows for more effective research assessment. You are encouraged to use the free text boxes beneath each contributing author's name to add specific details on the author's contribution. More information is available in our guide to authors:

<https://www.embopress.org/page/journal/17574684/authorguide#authorshipguidelines>

- Indicate in legends exact n and exact p values, not a range, along with the statistical test used. To keep the figures "clear" some authors found providing an Appendix table Sx with all exact p-values preferable. You are welcome to do this if you want to.

- In Methods, please include statement that the experiments with human samples conformed to the principles set out in the WMA Declaration of Helsinki and the Department of Health and Human Services Belmont Report.

- Please remove Reagents and Tools Table and uploaded it as a separate file. Structured Methods section includes Reagents and Tools Table followed by a Methods and Protocols section. More information on how to adhere to this format as well as downloadable templates (.docx) for the Reagents and Tools Table can be found in our author guidelines:

<https://www.embopress.org/page/journal/17574684/authorguide#structuredmethods>

An example of a paper with Structured Methods can be found here:

<https://www.embopress.org/doi/full/10.1038/s44320-024-00037-6#sec-4>

- Please use the following format to report the accession number of your data:

[data type]: [full name of the resource] [accession number/identifier] [(doi or URL or identifiers.org/DATABASE:ACCESSION)]

Please check "Author Guidelines" for more information.

<https://www.embopress.org/page/journal/17574684/authorguide#availabilityofpublishedmaterial>

4) Appendix: Please add page numbers to the table of content.

5) Synopsis: Every published paper now includes a 'Synopsis' to further enhance discoverability. Synopses are displayed on the journal webpage and are freely accessible to all readers. They include separate synopsis image and synopsis text.

- Synopsis image: Please provide a visual abstract as a high-resolution jpeg file 550 pixels wide x 300 - 600 pixels high to illustrate your article.

- Synopsis text: Please provide a short standfirst (maximum of 300 characters, including space) as well as 2-5 one sentence bullet points that summarise the paper as a .doc file. Please write the bullet points to summarise the key NEW findings. They should be designed to be complementary to the abstract - i.e. not repeat the same text. We encourage inclusion of key acronyms and quantitative information (maximum of 30 words / bullet point). Please use the passive voice.

6) As part of the EMBO Publications transparent editorial process initiative (see our Editorial at

<http://embomolmed.embopress.org/content/2/9/329>), EMBO Molecular Medicine will publish online a Review Process File (RPF) to accompany accepted manuscripts. This file will be published in conjunction with your paper and will include the anonymous referee reports, your point-by-point response and all pertinent correspondence relating to the manuscript. Let us know whether you agree with the publication of the RPF and as here, if you want to remove or not any figures from it prior to publication.

7) Please provide a point-by-point letter INCLUDING my comments as well as the reviewer's reports and your detailed responses (as Word file).

I look forward to reading a new revised version of your manuscript as soon as possible.

Yours sincerely,

Zeljko Durdevic

Zeljko Durdevic
Senior Editor
EMBO Molecular Medicine

*** Instructions to submit your revised manuscript ***

1) a .docx formatted version of the manuscript text (including Figure legends and tables)

2) Separate figure files*

3) supplemental information as Expanded View and/or Appendix. Please carefully check the authors guidelines for formatting Expanded view and Appendix figures and tables at <https://www.embopress.org/page/journal/17574684/authorguide#expandedview>

4) a letter INCLUDING the reviewer's reports and your detailed responses to their comments (as Word file).

5) The paper explained: EMBO Molecular Medicine articles are accompanied by a summary of the articles to emphasize the major findings in the paper and their medical implications for the non-specialist reader. Please provide a draft summary of your article highlighting

This may be edited to ensure that readers understand the significance and context of the research.

Please refer to any of our published articles for an example.

6) Author contributions: the contribution of every author must be detailed in a separate section.

7) EMBO Molecular Medicine now requires a complete author checklist

(<https://www.embopress.org/page/journal/17574684/authorguide>) to be submitted with all revised manuscripts. Please use the checklist as guideline for the sort of information we need WITHIN the manuscript. The checklist should only be filled with page numbers where the information can be found. This is particularly important for animal reporting, antibody dilutions (missing) and exact values and n that should be indicated instead of a range.

8) Every published paper now includes a 'Synopsis' to further enhance discoverability. Synopses are displayed on the journal webpage and are freely accessible to all readers. They include a short stand first (maximum of 300 characters, including space) as well as 2-5 one sentence bullet points that summarise the paper. Please write the bullet points to summarise the key NEW findings. They should be designed to be complementary to the abstract - i.e. not repeat the same text. We encourage inclusion of key acronyms and quantitative information (maximum of 30 words / bullet point). Please use the passive voice. Please attach these in a separate file or send them by email, we will incorporate them accordingly.

You are also welcome to suggest a striking image or visual abstract to illustrate your article. If you do please provide a jpeg file 550 px-wide x 300-600px high.

9) A Conflict of Interest statement should be provided in the main text

10) Please note that we now mandate that all corresponding authors list an ORCID digital identifier. This takes <90 seconds to complete. We encourage all authors to supply an ORCID identifier, which will be linked to their name for unambiguous name identification.

Currently, our records indicate that the ORCID for your account is 0000-0003-1120-030X.

Link Not Available

11) Include a Reagents and Tools Table as part of the Methods section, which can be downloaded from our author guidelines (<https://www.embopress.org/page/journal/17574684/authorguide#structuredmethods>)

Photos 400-800 DPI

*Additional important information regarding figures and illustrations can be found at <https://bit.ly/EMBOPressFigurePreparationGuideline>. See also figure legend preparation guidelines: <https://www.embopress.org/page/journal/17574684/authorguide#figureformat>

***** Reviewer's comments *****

Referee #2 (Remarks for Author):

good job

The authors addressed the remaining editorial issues.

20th Jun 2025

Dear Dr. Wang,

We are pleased to inform you that your manuscript is accepted for publication and is now being sent to our publisher to be included in the next available issue of EMBO Molecular Medicine.

Zeljko Durdevic
Senior Editor
EMBO Molecular Medicine
